# Seizures are a druggable mechanistic link between TBI and subsequent tauopathy

Hadeel Alyenbaawi[1,2,3], Richard Kanyo[1,4], Laszlo F Locskai[1,4], Razieh Kamali-Jamil[1,5], Michèle G DuVal[4], Qing Bai[6], Holger Wille[1,5], Edward A Burton[6,7], W Ted Allison[1,2,4]*

[1]Centre for Prions & Protein Folding Disease, University of Alberta, Edmonton, Canada; [2]Department of Medical Genetics, University of Alberta, Edmonton, Canada; [3]Majmaah University, Majmaah, Saudi Arabia; [4]Department of Biological Sciences, University of Alberta, Edmonton, Canada; [5]Department of Biochemistry, University of Alberta, Edmonton, Canada; [6]Department of Neurology, University of Pittsburgh, Pittsburgh, United States; [7]Geriatric Research, Education and Clinical Center, Pittsburgh VA Healthcare System, Pittsburgh, United States

**Abstract** Traumatic brain injury (TBI) is a prominent risk factor for dementias including tauopathies like chronic traumatic encephalopathy (CTE). The mechanisms that promote prion-like spreading of Tau aggregates after TBI are not fully understood, in part due to lack of tractable animal models. Here, we test the putative role of seizures in promoting the spread of tauopathy. We introduce 'tauopathy reporter' zebrafish expressing a genetically encoded fluorescent Tau biosensor that reliably reports accumulation of human Tau species when seeded via intraventricular brain injections. Subjecting zebrafish larvae to a novel TBI paradigm produced various TBI features including cell death, post–traumatic seizures, and Tau inclusions. Bath application of dynamin inhibitors or anticonvulsant drugs rescued TBI-induced tauopathy and cell death. These data suggest a role for seizure activity in the prion-like seeding and spreading of tauopathy following TBI. Further work is warranted regarding anti-convulsants that dampen post-traumatic seizures as a route to moderating subsequent tauopathy.

*For correspondence:
ted.allison@ualberta.ca

Competing interests: The authors declare that no competing interests exist.

## Introduction

Traumatic brain injury (TBI) is a leading cause of mortality and disability worldwide (*Hay et al., 2016*; *Nguyen et al., 2016*; *Rimel et al., 1981*). It is also a prominent risk factor for neurodegeneration and dementia, such as chronic traumatic encephalopathy (CTE) (*Chauhan, 2014*; *Gardner and Yaffe, 2015*; *Uryu et al., 2007*). TBI can result from direct physical insults, from rapid acceleration and deceleration of the brain, or from shock wave impacts such as pressure waves emanating from explosive blasts (*Cruz-Haces et al., 2017*). Regardless, the primary mechanisms have much in common and the neuropathology in TBI and CTE patients includes the wide distribution of hyperphosphorylated Tau pathology, axonal degeneration, and neuronal loss (*Hay et al., 2016*; *Johnson et al., 2013*; *McKee et al., 2015*; *Ojo et al., 2016*). The mechanisms whereby physical injury is translated into progressive Tau pathology remain unresolved and represent prospective therapeutic targets. Progress on this front is hampered by lack of access to suitable models: applying physical injury to a cell culture is difficult and poorly represents the complex biopathology that intertwines many multifaceted aspects of brain physiology.

The progressive deposition of hyperphosphorylated Tau protein in filamentous forms is a defining hallmark of tauopathies, which includes Alzheimer's disease (AD), CTE, and several other dementias. Each of the tauopathies affects distinct brain regions and has a unique clinical presentation (*Kovacs, 2017*; *Orr et al., 2017*). Early in CTE, hyperphosphorylated Tau is accumulated in a cluster

**eLife digest** Traumatic brain injury can result from direct head concussions, rapid head movements, or a blast wave generated by an explosion. Traumatic brain injury often causes seizures in the short term and is a risk factor for certain dementias, including Alzheimer's disease and chronic traumatic encephalopathy in the long term. A protein called Tau undergoes a series of chemical changes in these dementias that makes it accumulate, form toxic filaments and kill neurons.

The toxic abnormal Tau proteins are initially found only in certain regions of the brain, but they spread as the disease progresses. Previous studies in Alzheimer's disease and other diseases where Tau proteins are abnormal suggest that Tau can spread between neighboring neurons and this can be promoted by neuron activity. However, scientists do not know whether similar mechanisms are at work following traumatic brain injury. Given that seizures are very common following traumatic brain injury, could they be partly responsible for promoting dementia?

To investigate this, researchers need animal models in which they can measure neural activity associated with traumatic brain injury and observe the spread of abnormal Tau proteins. Alyenbaawi et al. engineered zebrafish so that their Tau proteins would be fluorescent, making it possible to track the accumulation of aggregated Tau protein in the brain. Next, they invented a simple way to perform traumatic brain injury on zebrafish larvae by using a syringe to produce a pressure wave. After this procedure, many of the fish exhibited features consistent with progression towards dementia, and seizure-like behaviors.

The results showed that post-traumatic seizures are linked to the spread of aggregates of abnormal Tau following traumatic brain injury. Alyenbaawi et al. also found that anticonvulsant drugs can lower the levels of abnormal Tau proteins in neurons, preventing cell death, and could potentially ameliorate dementias associated with traumatic brain injury. These drugs are already being used to prevent post-traumatic epilepsy, but more research is needed to confirm whether they reduce the risk or severity of Tau-related neurodegeneration.

of perivascular neurons and glia in the depths of cortical sulci. Later in CTE, Tau pathology is widespread and incorporates cortical and subcortical gray-matter areas (*Hay et al., 2016*; *Johnson et al., 2012*; *McKee et al., 2015*). This broad spreading of Tau pathology in CTE can also be observed following TBI ascribed to single trauma events (*Johnson et al., 2012*). This spreading of Tauopathy is consistent with a prion-like mechanism; indeed, brain homogenates from mice subjected to TBI can initiate p-Tau pathology when injected into healthy wild-type mice (*Zanier et al., 2018*). The recipient mice develop a p-Tau pathology similar to single severe TBI patients, which then spreads from injection sites to distant regions, behaving similarly to bona fide prions (*Zanier et al., 2018*).

Beyond TBI, the self-propagation and prion-like spread of Tau aggregates is thought to play a key role in the progression of other tauopathies such as AD (*Iba et al., 2013*; *Iba et al., 2015*; *Mudher et al., 2017*; *Narasimhan et al., 2017*; *Sanders et al., 2014*). Mechanisms of Tau spreading, and the therapeutic targets they offer, have principally been defined in vitro and include tunnelling nanotubes and extracellular vesicles (EVs such as exosomes and synaptic vesicles) and their uptake via endocytosis (*Colin et al., 2020*; *Demaegd et al., 2018*; *Evans et al., 2018*). In AD and other tauopathies, observations from patients and mice have highlighted the capacity of Tau seeds to spread trans-synaptically (*Goedert et al., 1989*; *Pickett et al., 2017*). Moreover, it has been shown that neuronal activity serves an important role in the spread of Tau pathology and general proteostasis (*Pickett et al., 2017*; *Wu et al., 2016*; *Yamada et al., 2014*). Stimulation of neuronal activity increased the extracellular release of Tau to the media in vitro and enhanced Tau pathology in a mouse model of familial frontotemporal dementia (*Pickett et al., 2017*; *Wu et al., 2016*; *Yamada et al., 2014*). Whether similar mechanisms of Tau release and spread occur following TBI remains unknown.

In this light, an intriguing aspect of TBI is the prominence of post-traumatic seizures that might be predicted to initiate the aggregation and/or exacerbate the spread of Tau pathology. Seizures are one of the key consequences of all types of TBI, and they have been more commonly reported in patients who suffered from blast injuries (*Asikainen et al., 1999*; *Salinsky et al., 2015*). Although

the exact prevalence remains undetermined (*Lucke-Wold et al., 2015*), it is anticipated that over 50% of TBI patients with severe injuries develop seizures or post-traumatic epilepsy (*Kovacs et al., 2014*). A link between seizures and Tau pathology is suggested by increased prevalence of seizures in AD patients and animal models of AD (*Sánchez et al., 2018*; *Yan et al., 2012*). Whether reducing post-traumatic seizures can delay or minimize the progression of tauopathy has yet to be fully explored.

This knowledge gap is due in part to a lack of accessible in vivo models that can report the progression and spread of tauopathy, or that allow neural activity associated with TBI to be measured and manipulated. To address these issues, we engineered a tauopathy biosensor transgenic zebrafish that develops GFP+ puncta when Tau aggregates within the brain or spinal cord. Additionally, we introduce a simple medium-throughput method to induce TBI in zebrafish larvae. Combining these novel approaches, we found that post-traumatic seizures correlate strongly with spreading tau pathology following TBI. Manipulating this seizure activity mitigated Tau aggregation and revealed a critical role for endocytosis in the prion-like spread of Tau seeds in vivo following TBI. The results from our novel in vivo TBI model implicate seizures and dynamin-dependent endocytosis in the spread of Tau seeds, thereby offering potential therapeutic targets.

## Results

### Engineering and validation of tauopathy reporter lines

Previous reports describe the assessment and quantification of Tau inclusions in living cells (typically Human Embryonic Kidney cells), via measuring aggregation of fluorescent proteins fused to Tau protein, providing sensitive detection of pathological Tau species and strain variants (*Kaufman et al., 2016*; *Sanders et al., 2014*; *Woerman et al., 2016*). Tau is predominantly expressed in the neurons of the CNS, and we reasoned that fluorescent biosensor tools would have good potential to reveal additional phenotypes when expressed in these cells and moreover, that prion-like mechanisms of tauopathy spread are best modeled in an intact brain (e.g. vectored by blood and glymphatic circulation, ventricles, axonal projections, and immune systems). Therefore, we engineered a tauopathy biosensor transgenic zebrafish that expresses a fluorescent Tau reporter protein. Our genetically encoded fluorescent reporter protein was composed of the sequence of the human Tau core-repeat domain fused to green fluorescent protein (GFP) with a linker sequence and is referred to here as Tau4R-GFP (*Figure 1A* and its *Figure 1—figure supplement 1A*). Contrasting previous in vitro models, our biosensor did not feature any pro-aggregation mutations in the human Tau repeats; this design was intended to minimize spontaneous aggregation events. The expression of the biosensor protein in zebrafish was under the control of the pan-neuronal promoter *neuronal enolase 2* (*eno2*, see *Bai et al., 2007*), which drives expression throughout the CNS (*Figure 1B* and its *Figure 1—figure supplement 1C*). We deployed the transgene in a transparent zebrafish line (the 'Casper' background [*White et al., 2008*]) to facilitate analysis beyond the early larval development stages (when pigmentation would otherwise begin to obscure microscopy). We isolated a stable transgenic (Tg) line that expresses the Tau4R-GFP biosensor reporter robustly and clearly in the CNS (*Figure 1B*), *Tg(eno2:Hsa.MAPT_Q244-E372−EGFP)$^{ua3171}$*, and assigned it allele number ua3171.

Simultaneously, we expressed the same biosensor in vitro to validate the construct we deployed in vivo (*Figure 1—figure supplement 1A and B*). Both in HEK293T cells and Tg zebrafish, immunoblotting using anti-GFP antibody detected our Tau-4R-GFP reporter protein at the expected size of ~45 Kd, similar to a SOD1:GFP biosensor protein of similar predicted size, and an appropriately larger size relative to GFP protein alone (*Figure 1C* and its *Figure 1—figure supplement 1B'*).

We assessed the capacity of our Tau4R-GFP biosensor to report the presence of Tau pathology via transducing brain homogenates into cells. Brain homogenates burdened with tauopathy, from transgenic mice expressing mutant human Tau (Tg Tau$^{P301L}$), were compared to normal non-Tg mouse homogenates as a negative control. Congruent with findings obtained in past similar cell assays (*Sanders et al., 2014*), GFP-positive (GFP+) inclusions were detected only when cells were transduced with brain homogenate containing pathogenic human Tau fibrils (from Tg Tau$^{P310L}$ mice) (*Figure 1—figure supplement 1D*). The in vitro assay detection rate was approximately 5% of cells having GFP+ inclusions in total, with 2% of cells forming multiple nuclear puncta and ~3% forming one cytoplasmic inclusion, whereas various negative controls consistently displayed 0% of cells with

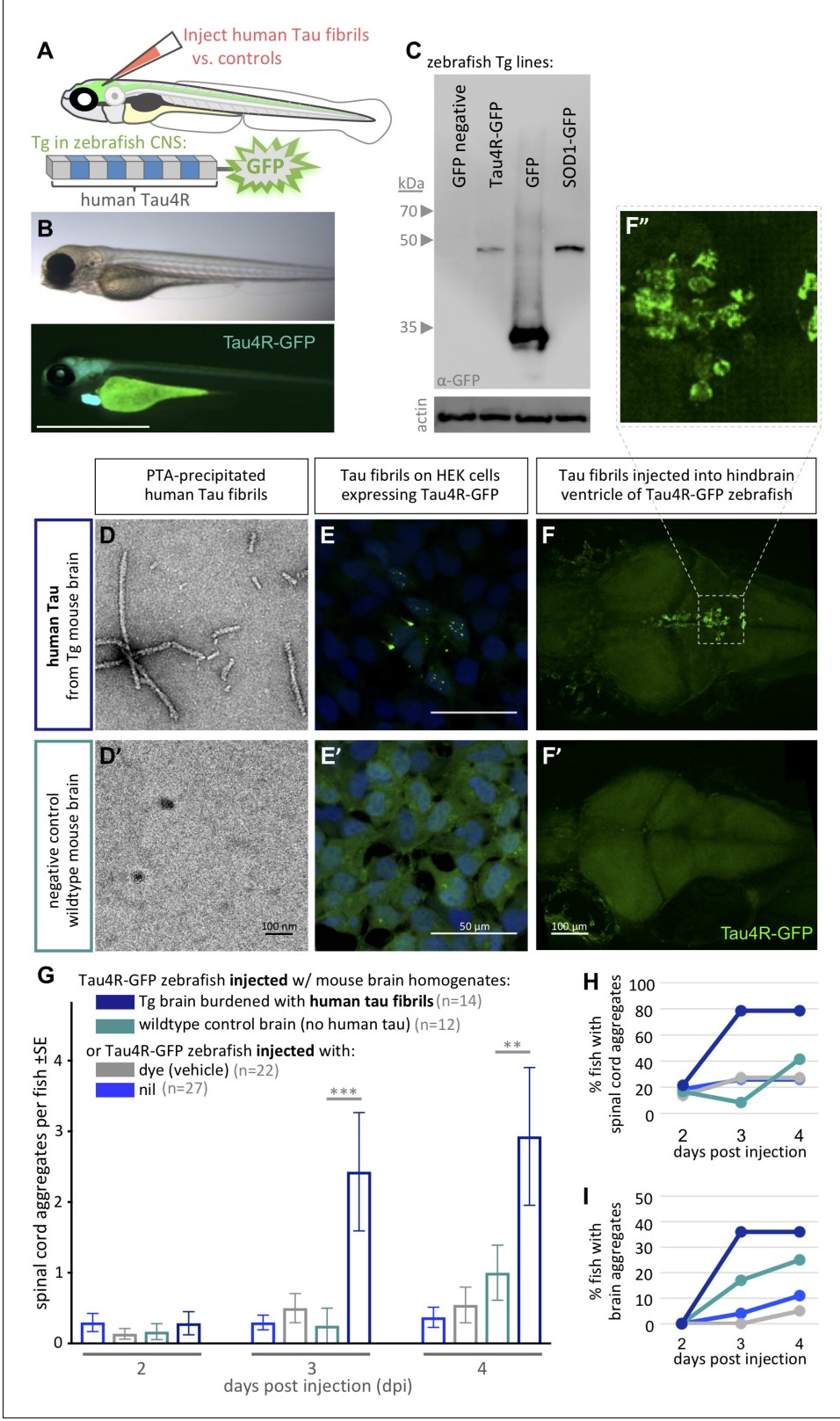

**Figure 1.** Validating tauopathy fluorescent biosensor in vitro and in zebrafish. The biosensor Tau4R-GFP was validated for its ability to detect tauopathy seeds in vitro and in zebrafish. (**A**) Schematic of Tau4R-GFP 'Tau biosensor' that contains the four binding repeats (4R) region of wild-type human Tau linked to green fluorescent protein (GFP; see also *Figure 1—figure supplement 1A*). (**B**) Transgenic zebrafish engineered to express Tau4R-

*Figure 1 continued on next page*

*Figure 1 continued*

GFP biosensor throughout neurons of the CNS. Wild-type GFP is also abundant in the heart, which serves as a marker of the transgene being present but is otherwise irrelevant to our analyses. Scale bar ≅1 mm. (**C**) Western blot on zebrafish brain confirmed production of Tau4R-GFP at the expected size, similar to a SOD1-GFP biosensor and coordinately larger than GFP alone. (**D**) Human Tau fibril precipitated from transgenic (Tg Tau^P301L) mouse brain homogenates using PTA and assessed by EM. (**E**) Application of PTA-purified brain homogenate induced the formation of Tau inclusions similar to clarified brain homogenate (scale bar 50 μm; compare to *Figure 1—figure supplement 1D*), but application of equivalent preparations from non-Tg mice produced no GFP+ inclusions. (**F–I**) Tau biosensor zebrafish detects disease-associated human Tau fibrils following intraventricular injection of brain homogenate. Crude brain homogenates were microinjected into the hindbrain ventricle of Tau4R-GFP zebrafish larvae at 2 days post-fertilization, and Tau inclusions were analyzed at several time points. (**F**) Tau biosensor zebrafish larvae developed readily apparent GFP+ inclusions in the brain and spinal cord (*Figure 1—figure supplement 2*) when injected with brain homogenate burdened with Tau pathology (from Tg mice) but not from healthy brain homogenate (**F'**, from non-Tg mice). **F'** inset shows many adjacent cells exhibiting GFP+ Tau aggregates. (**G**) Tau biosensor zebrafish injected with human Tau fibrils (within Tg mouse brain homogenate) developed significantly more aggregates on the spinal cord compared to uninjected control and other control groups, including compared to wildtype mouse brain homogenate (**p=0.0033 ***p=0.0006, ordinary two-way ANOVA and Tukey's multiple comparison test). (**H**) Same data as in G, expressed as the percentage of larval fish showing Tau aggregates in the spinal cord, and (**I**) those same fish also showed Tau aggregates in the brain, over time. n = number of individual larvae. Images in E and F are 5 days post-application or post-injection, respectively.

The online version of this article includes the following figure supplement(s) for figure 1:

**Figure supplement 1.** Quantification GFP+ inclusions in HEK cells expressing Tau4R-GFP biosensor.

**Figure supplement 2.** Quantification GFP+ inclusions in CNS of zebrafish expressing Tau4R-GFP biosensor, following injection of human Tau into the zebrafish hindbrain ventricle.

**Figure supplement 3.** Movement of some Tau puncta over time following injection of zebrafish larvae with brain homogenate burdened with human tauopathy.

---

inclusions (*Figure 1—figure supplement 1E*). To verify that Tau aggregates in the clarified brain homogenate caused the GFP+ puncta, we purified Tau aggregates from the tissue samples using PTA precipitations (*Woerman et al., 2016*). Tau fibrils purified from these preparations were characterized via EM analysis (*Figure 1D*). Transducing these preparations (in contrast to control preparations derived from non-Tg mice) produced fluorescent puncta in the Tau4R-GFP reporter cells (*Figure 1E*), confirming the ability of our Tau4R-GFP chimeric protein to report Tau aggregation.

## Validation of in vivo Tau biosensor via intraventricular brain injections of Tau fibrils

To test if the Tau4R-GFP biosensor can report the in vivo progression of tauopathy, we emulated intracerebral injection methods that induce (prion-like) Tau pathology in mice (*Clavaguera et al., 2013*; *Guo et al., 2016*; *Peeraer et al., 2015*). We injected clarified brain homogenate laden with human Tau fibrils, prepared as above from Tg mice, into the hindbrain ventricle of 2 days post-fertilization (dpf) Tau biosensor zebrafish (*Figure 1—figure supplement 2A*). The injected larvae and control groups were monitored daily for up to 4 days post-injection (dpi). Biosensor larvae injected with human Tau fibrils (from Tg mouse brain) developed GFP+ puncta, reflective of Tau aggregation in the brain (*Figure 1F,F''*). These Tau inclusions were prominent near the ventricle wall as well as in sensory neurons along the spinal cord, when injected with brain homogenate from human-tau transgenic mouse (*Figure 1—figure supplement 2B*). These puncta sometimes appeared to have either a lone dot-like shape or were similar to the multiple nuclear puncta detected in vitro, in which three to four small puncta are clustered together, or in other instances were more diffuse and concentrated outside the nucleus (*Figure 1F,F''*). Repeated assessment of the location of Tau aggregates on the spinal cord of the same individuals over multiple days, using somite numbers as landmarks, suggested a movement of some of these puncta over time (*Figure 1—figure supplement 3*).

The abundance of GFP+ spinal cord inclusions was progressive and significantly higher in larvae injected with pathogenic Tau brain homogenate compared with various controls (*Figure 1G*. p=0.0006 and p=0.0033 at 3dpi and 4dpi, respectively compared to injection of healthy wild-type control brain, ordinary two-way ANOVA and Tukey's multiple comparison test). Few larvae in the

control groups developed spontaneous inclusions but the number of the larvae and the abundance of those inclusions were minimal (*Figure 1G*). A total of 80% and 35% of the larvae injected with human Tau fibrils developed puncta in the brain and spinal cord, respectively (*Figure 2H and I*). On the other hand, a lower proportion of TAu biosensor larvae developed 'spontaneous' inclusions post-injection with the control brain homogenate from wild-type mice (*Figure 2H and I*). Tau aggregates were detected on the spinal cord region as early as 2 dpi. Intriguingly, a small percentage of larvae developed sporadic GFP+ Tau aggregates regardless of treatment. Visualizing the data as distributions of larvae with particular abundances of GFP+ inclusions (*Figure 1—figure supplement 2C*) highlights a trend where most larvae did not develop aggregates unless they were injected with brain homogenate containing fibrillar, pathogenic human Tau species. In those cases, the biosensor larvae developed an abundant number of aggregates. Overall, these data confirm the ability of our biosensor model to detect pathogenic Tau species in vivo.

Like other protein misfolding diseases, tauopathies reflect a proteostatic imbalance wherein the clearance of pathological Tau species is insufficient relative to accumulation (*Chiti and Dobson, 2006*; *Lim and Yue, 2015*). We reasoned that if the Tau biosensor larvae are faithfully reflecting Tau proteostasis concepts in vivo, then this could be revealed via inhibition of the proteasome with MG-132. Larvae treated with MG-132 had GFP+ puncta in their brains (*Figure 2A*) at a rate approximately double to the occurrence of spontaneous GFP+ inclusions (*Figure 2B*). Following injection of mouse brain homogenate containing human Tau fibrils, applying the proteasome inhibitor MG-132 substantially enhanced the percentage of larvae bearing Tau4R-GFP+ inclusions in the brain (to ~70%, *Figure 2B*), relative to equivalent larvae without MG-132 (~36%, *Figure 2B*).

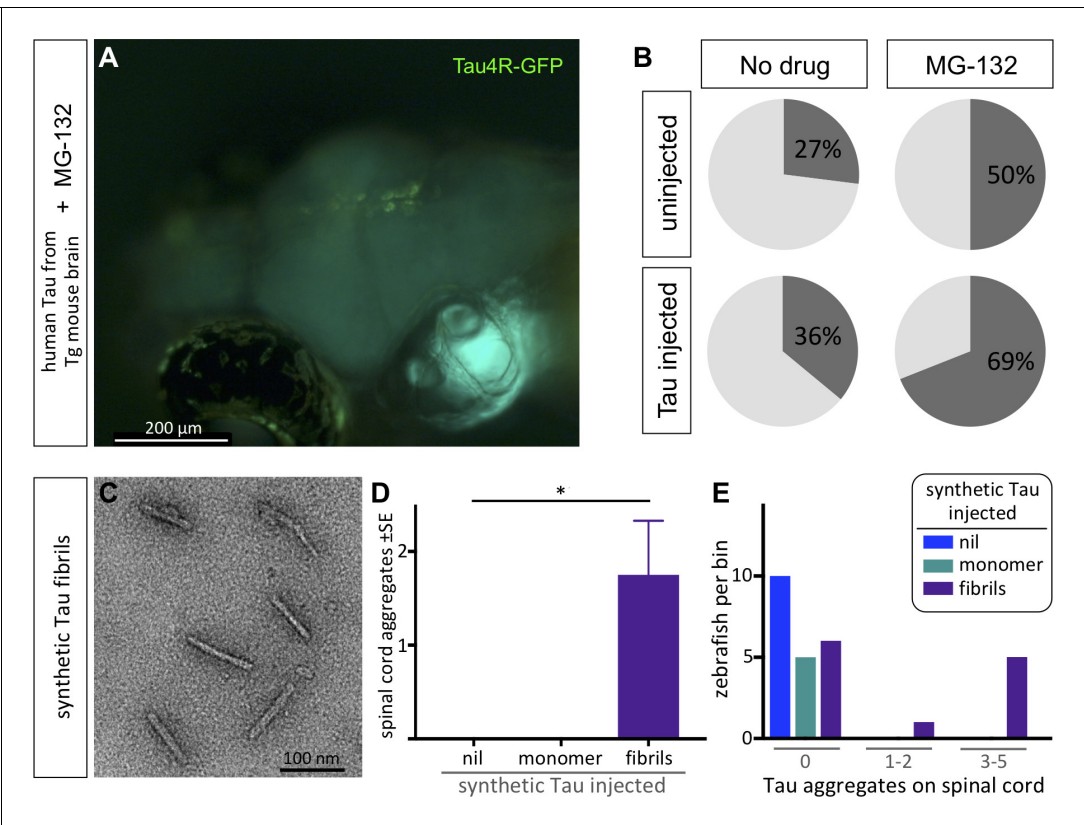

**Figure 2.** Protein-only induction of Tau puncta in vivo detected in biosensor zebrafish. Injections of synthetic Tau fibrils into Tau4R-GFP zebrafish induced GFP+ puncta in brains and spinal cord. (A,B) Inhibiting the proteasome with MG-132 enhanced the percentage of larvae bearing GFP+ inclusions in the brain following injection of tau-laden brain homogenate. (C) Synthetic human Tau proteins were fibrillized as confirmed via EM analysis. Human Tau fibrils were microinjected into the larval hindbrain at 2 days post-fertilization, and Tau inclusions were analyzed at 3 days post injections. (D) Tau aggregates were only observed after injection of Tau fibrils, not monomers (*p=0.0104, Kruskal Wallis test). (E) Tau aggregates (same data as D, presented as distribution of larvae that displayed various amounts of GFP+ puncta) appear only after injection of Tau fibrils, not monomers.

It was striking that the zebrafish Tau biosensor was robustly able to discriminate brain homogenates that were burdened with human Tau aggregates versus those that were not. However, we considered an alternative explanation for the data: the difference may not depend directly on human Tau in the brain homogenate but could instead reflect other bioactive components of the degenerating Tg mouse brain. To verify that the formation of GFP+ puncta in zebrafish can be seeded by a protein-only injection, we delivered synthetic human Tau protein (2N4R). After confirming the recombinant Tau proteins were appropriately fibrillized via EM (*Figure 2C*), we delivered them by intraventricular injections as described above. Similar to previous data with brain homogenate, the larvae that were injected with synthetic Tau fibrils developed inclusions proximal to the brain ventricles as well as along the spinal cord at 3–6 dpi. The abundance of Tau aggregates along the spinal cord was significantly higher in larvae injected with the synthetic Tau fibrils compared to larvae injected with Tau monomers or to the non-injected group (p=0.0104) (*Figure 2D*). The distribution of larvae based on the number of Tau aggregates they accumulated also supported these findings (*Figure 2E*). In sum, the Tau4R biosensor deployed in the CNS of larval zebrafish was able to report Tau species, and further revealed the prion-like induction of tauopathy via protein-only seeding in vivo.

## Introduction of the first TBI model for larval zebrafish

We next sought to deploy our Tau biosensor in a tauopathy model that enables higher throughput than can be achieved with intraventricular injection methods. We considered TBI) as an inducer of the tauopathy in CTE; further, we were encouraged that innovations in this realm could fill an unmet need for a high-throughput, genetically tractable in vivo model of these devastating concussive injuries. Although a few methods have been reported to induce TBI in *adult* zebrafish that are comparable to mammalian TBI methods (*Maheras et al., 2018*; *McCutcheon et al., 2017*), no such methods were available for zebrafish larvae (although *McCutcheon et al., 2016* argue their application of exogenous glutamate may help address aspects of excitotoxicity associated with such insults). Here, we introduce and validate a simple and inexpensive method to induce TBI in zebrafish larvae. Investigating TBI in larvae offers substantial benefits regarding experimental throughput, economy, accessibility of drug and genetic interventions, and bioethics.

We devised a traumatic injury paradigm by loading zebrafish larvae (~12 individuals in their typical E3 liquid growth media) into a syringe with a closed valve stopper, and applying a hit on the plunger to produce a pressure wave through the fish body akin to pressure or shock waves experienced during human blast injury (*Nakagawa et al., 2011*; *Figure 3A*). To challenge the method's reproducibility, and to permit manipulation of injury intensity, a series of defined masses were dropped on the syringe plunger. Technical variability, anticipated from larvae being in different orientations and positions within the syringe, was reduced by applying the injury three times to each group of larvae (except where noted otherwise) while repositioning the syringe between each injury. To assess if our method faithfully induced TBI similar to injury from pressure waves, we examined multiple markers known to be associated with blast-induced TBI, including cell death, hemorrhage, blood flow abnormalities, and tauopathy (*Bir et al., 2012*; *Kovacs et al., 2014*; *Nakagawa et al., 2011*). Additionally, we evaluated the occurrence of post-traumatic seizure activity and increases in neuronal activity acutely associated with the trauma.

We established the TBI method via empirical testing of various parameters, restricting ourselves to materials and methods that can be adopted inexpensively, with a goal of consistently inducing a robust injury (see phenotypes below) vs. a tradeoff with maximizing survival of the larvae. Subsequent to this optimization, we were able to characterize the pressure induced within the syringe during each injury (*Figure 3B–D*). The maximum pressure induced was near 170 kPa (*Figure 3B*). The dynamics of the pressure change events during TBI (*Figure 3B*) imply that dropping the heavier weights led to the weight bouncing and producing a secondary increase in pressure (e.g. at ~175 or ~275 ms in *Figure 3B*). The maximal pressure induced varied from ~130 to ~175 kPa in an approximately linear fashion depending on the mass of the weight dropped (*Figure 3D*). The mean pressure change over the first 300 ms of the TBI also increased in a nearly linear fashion, and increased by nearly an order of magnitude when dropping weights of 30 g compared to 300 g (*Figure 3C*).

We evaluated TBI-induced hemorrhage via the use of *Tg(gata1a:DsRed)* larvae that have red fluorescence in their blood cells (*Traver et al., 2003*). Hemorrhage was observed variably in larvae when a heavy weight (300 g) was used to induce the traumatic injury (*Figure 3E*). Further, approximately

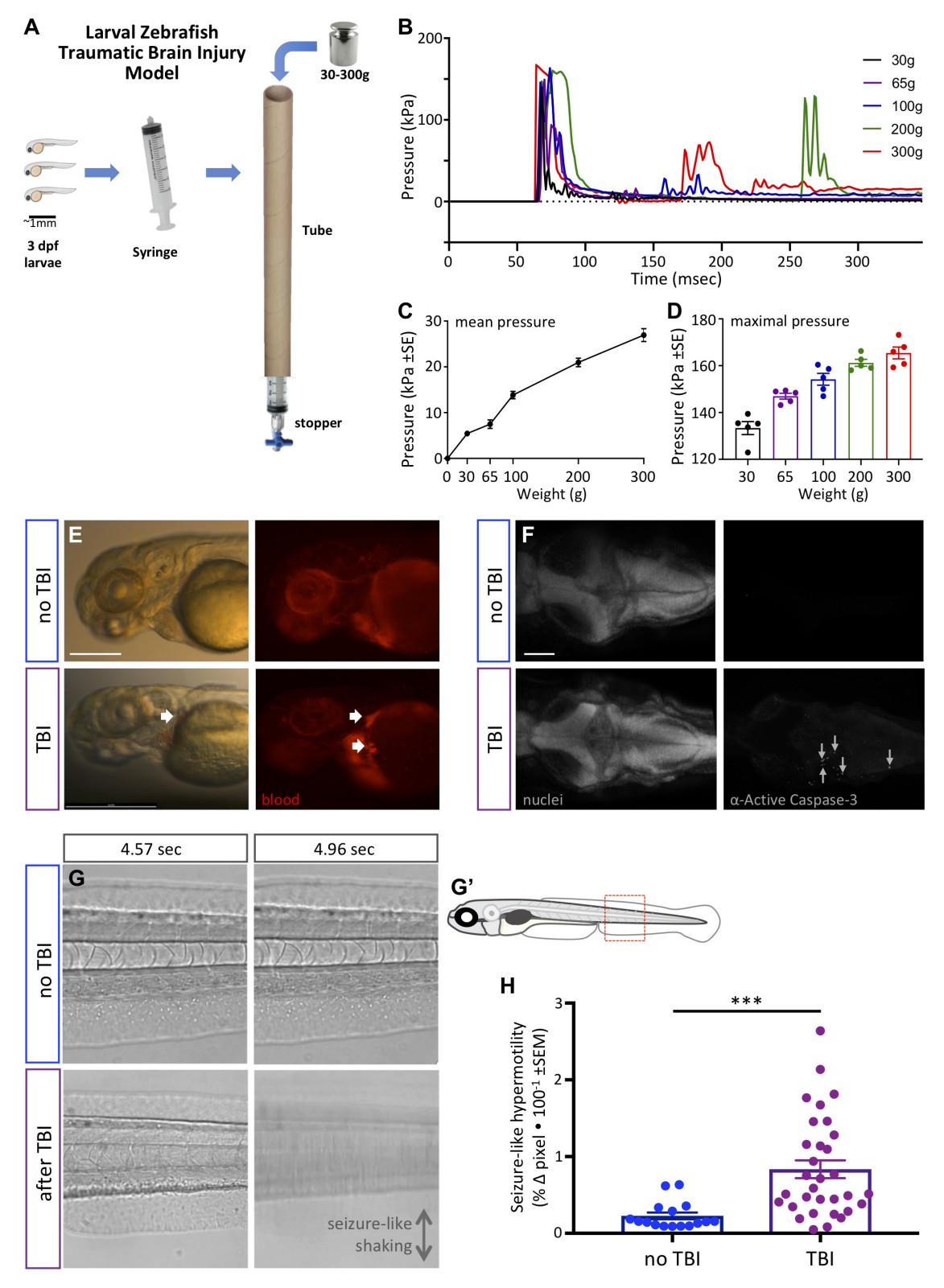

**Figure 3.** Zebrafish larvae subjected to traumatic brain injury (TBI) exhibited various biomarkers of TBI. (**A**) A novel TBI model for larval zebrafish: to induce blast injury, zebrafish larvae were loaded into a syringe with a stopper. A defined weight was dropped on the syringe plunger from a defined height, producing a pressure wave through the fish body akin to pressure waves experienced during human blast injury. (**B**) Dynamics of the pressure increase after dropping weights of varying masses in our TBI model. (**C,D**) The mean and maximum pressures generated, respectively, by various

*Figure 3 continued on next page*

*Figure 3 continued*

weights applied in the TBI model. Dots represent individual trials. (**E**) Hemorrhage after TBI was observed in some of the larvae fish using *Tg[gata1a: DsRed]* transgenic zebrafish that express DsRed in erythrocytes, as indicated by white arrows. Lateral view of larval heads with anterior at the left. Scale bar ≅250 µm. (**F**) Increased cell death in the brain of 4 dpf larvae subjected to TBI as indicated by immunostaining of activated Caspase-3 (magenta). Positive and negative controls for immunostaining are in supplement. Nuclei were stained with DAPI in gray for reference. These are dorsal views of larval zebrafish brains with anterior at the left. Scale bar is 100 µm. (**G**) Seizure-like clonic shaking is observed in a subset of larvae after TBI. Movie frames are displayed from *Video 2*. These frames (left and right panels) are separated by ~400 ms in time, and are lateral views of the larval zebrafish trunk (akin to red box in G'). Control fish without TBI show little movement except obvious blood flow. Following TBI, larvae show bouts of calm (bottom left) interspersed (~400 ms later) with bouts of intense seizure-like convulsions (Stage III seizures; bottom right). (**H**) Larvae subjected to TBI also displayed Stage II seizures, that is weaker seizures that manifest as hypermotility and are detected using a previously optimized behavioral tracking software system – seizures are significantly more intense following TBI compared to the control group (***p=0.0013, paired t-test; dots are raw data for each larva, mean is plotted ± SE).

The online version of this article includes the following figure supplement(s) for figure 3:

**Figure supplement 1.** Traumatic brain injury (TBI)-induced cell death.

half of the TBI larvae showed abnormalities in blood flow including a temporary reduction or complete absence of blood circulation (*Video 1*), consistent with abnormalities detected in rodent TBI models (*Bir et al., 2012*). Subsequently, we assessed apoptosis in the TBI larvae, observing that our TBI method induced cell death in larvae as detected by staining for active Caspase-3 (*Figure 3F* and its *Figure 3—figure supplement 1*). The number of active-Caspase-3-positive cells was negligible in the control groups compared to a mean of 62 apoptotic cells in TBI larvae (SEM ±9.17, n = 3) and 75 (SEM ±4, n = 2) in positive-control-larvae (cell death induced with camptothecin, CPT; *Figure 3—figure supplement 1*). These data all align well with existing animal models of TBI with respect to mimicking characteristic features of human TBI, and support the effectiveness of our method in inducing TBI in larval zebrafish.

## TBI-treated larvae exhibited post-traumatic seizure-like behavior and increased neuronal activity during trauma

Post-traumatic seizures are one of the most frequent conditions associated with traumatic brain injuries and, despite being prevalent, remain poorly understood in TBI patients (*Kovacs et al., 2014*). Post-traumatic seizures were overtly apparent in a subset (approximately 40%) of zebrafish larvae after they were subjected to TBI. In some instances, the activity was highly reminiscent of Stage III seizures (defined previously in larval zebrafish as the most intense seizures; *Liu and Baraban, 2019*) with bouts of intense clonic convulsions and arrhythmic shaking (*Video 2*; exemplar frames from the movie are in *Figure 3G*). Other individuals exhibited hypermotility that is exactly consistent with past definitions of less intense Stage I or Stage II seizures. We quantified the latter seizure activity via behavioral tracking software (which we had previously optimized and validated for quantifying seizures in larval zebrafish [*Kanyo et al., 2020a*; *Leighton et al., 2018*]) and determined that larvae subjected to TBI exhibited seizure-like activity that was significantly higher than the control group (p=0.0013) (*Figure 3H*).

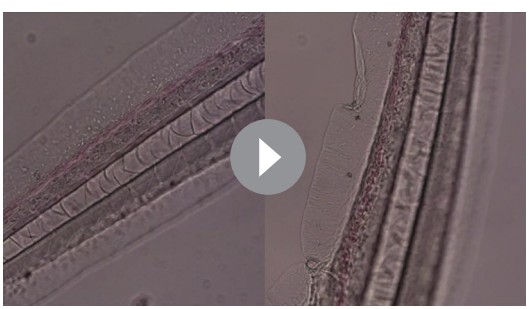

**Video 1.** TBI-induced blood flow abnormalties.
https://elifesciences.org/articles/58744#video1

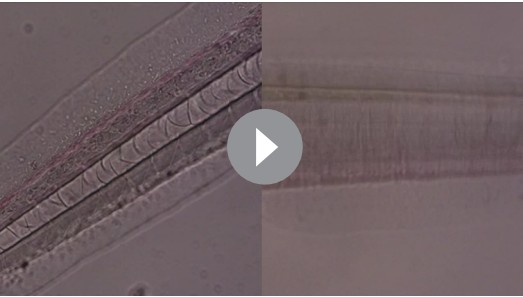

**Video 2.** TBI-induced seizures and blood flow abnormalities.
https://elifesciences.org/articles/58744#video2

Seizures are caused by abnormal and excessive neuronal excitability (*Stafstrom and Carmant, 2015*). To document bursts of neuronal activity *during* the brain trauma, if any, we utilized a genetically encoded calcium imaging CaMPARI reporter (calcium modulated photoactivatable ratiometric integrator) expressed throughout the CNS. CaMPARI fluoresces green in baseline conditions, and permanently converts to red fluorescent emission if high intracellular calcium levels (i.e. neural activity) occur coincident with application of 'photoconverting' intense 405 nm blue light. We subjected our allele of *Tg(elavl3:CaMPARI)*<sup>ua3144</sup> larvae (*Kanyo et al., 2020b*) to TBI, coincident with brief application of photoconverting light (405 nm light provided by an LED array directed at the syringe, as described in *Figure 4A*). A sharp increase in neuronal activity during TBI was evident, especially in the hindbrain region as indicated by enhanced red emission (*Figure 4B*). CaMPARI allows robust quantification of neural activity expressed as a ratio of red:green fluorescent emission, which confirmed that neuronal excitability increases significantly in response to brain trauma (*Figure 4C and D*). Notably, this combination of newly introduced methods of TBI being integrated with CaMPARI optogenetic methods (where the stable/irreversible changes from green to red fluorescent reportage allows a ratiometric quantification in a subsequent microscopy session) offers the rare ability to assess neural activity on un-restrained (free-swimming) subjects during TBI. In sum, our data reveal a substantial burst of neural activity occurs *during* TBI, and that zebrafish larvae exposed to TBI subsequently exhibit a significantly higher propensity for spontaneous seizures.

## TBI on tau-biosensor-zebrafish-larvae induced GFP+ puncta

After validating that our method was able to induce TBI upon zebrafish larvae, we next asked whether TBI induces Tau aggregates in our Tau biosensor model. Initially, we evaluated if our TBI method would induce aggregation of fluorescent proteins in models expressing GFP alone or other biosensor proteins such as SOD1-GFP (that is also designed to report prion-like protein aggregation). Following TBI, and regardless of injury intensity, no GFP+ aggregates were detected in these controls (*Figure 5—figure supplement 1*). Similar results were obtained with other transgenic zebrafish that express GFP in motor neurons (data not shown). Further, our Tau4R-GFP fish additionally express an unmodified GFP variant in the active heart muscle, and this robust GFP showed no sign of aggregation following TBI. Remarkably, in these same individual Tau4R-GFP larvae we detected Tau4R-GFP biosensor GFP+ puncta in both brains and spinal cords following TBI (*Figure 5A–B*). The abundance of GFP+ puncta increased with time following the injury (*Figure 5C–D*). To determine if the severity of tauopathy varies coordinately with severity of the traumatic injury, we assessed the impact of different masses. Although some variability is evident, a dose-response relationship is apparent such that the 65 g, 100 g, and 300 g weights induced more Tau aggregates compared to the control and 30 g weight (*Figure 5—figure supplement 4*). The heaviest weight (300 g) induced significantly more Tau aggregates versus the control group or the group with the 30 g weight (p<0.01 and p<0.001, respectively). Therefore, we decided to use both the 300 and 65 g weights for subsequent experiments. We evaluated whether dropping the light weight once or multiple times would affect the number of Tau aggregates on the spinal cord as well as dropping the weight once on 3 consecutive days, perhaps reminiscent of repetitive sports injury. We observed an increase in the abundance of Tau aggregates when the weight was dropped multiple times during 1 day, or over 3 consecutive days, but this increase was not statistically significant (*Figure 5—figure supplement 4*).

The GFP+ Tau aggregates formed in the brain region following TBI tend to form fused shapes (*Figure 5—figure supplement 4A*) reminiscent of the spontaneous aggregates described above. The aggregates on the spinal cord, however, had similar shapes to aggregates detected post brain-injections, but with qualitatively less brightness in some instances. Multi-day monitoring of individual larvae (*Figure 5—figure supplements 2* and *3*) revealed variation in formation of Tau aggregates amongst individual TBI larvae. We monitored the abundance of Tau aggregates within individual fish over time following TBI and found that the average tauopathy significantly increased compared to the control group (p=0.0224 at 3dpti and *p=0.0312 at 4dpti (days post-traumatic injury), *Figure 5C,D*). Analysis of distribution of larvae binned into the number of Tau4R-GFP+ puncta at 3 dpti showed that more larvae developed Tau4R-GFP+ puncta compared to the control group (inset in *Figure 5D*). Considering that many of the larvae subjected to TBI formed Tau4R-GFP+ puncta in the brain that had a fused pattern (*Figure 5—figure supplement 4A*), we focused on Tau

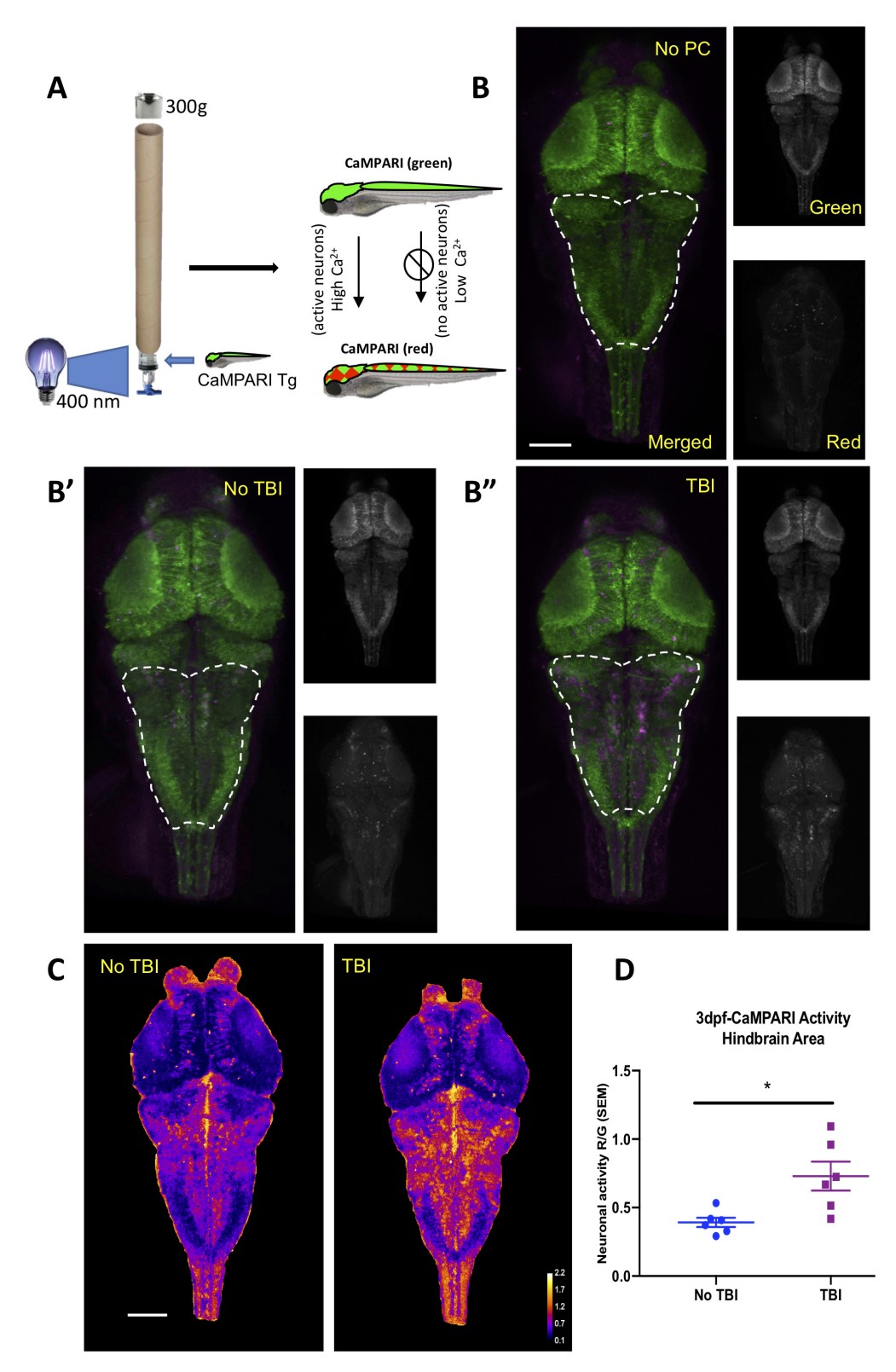

**Figure 4.** Neural activity increases during traumatic brain injury (TBI) as measured in CaMPARI zebrafish larva. (**A**) Schematic of TBI using CaMPARI (Calcium Modulated Photoactivatable Ratiometric Integrator) to optogenetically quantify neuronal excitability. Three dpf CaMPARI larvae were freely swimming while subjected to TBI, coincident with exposure to 405 nm photoconversion light. CaMPARI fluorescence permanently photoconverts from green to red emission only if the photoconversion light is applied while neurons are active (high intracellular [Ca$^{2+}$]). The ratio of red:green emission is

*Figure 4 continued on next page*

Figure 4 continued

stable such that it is quantifiable via subsequent microscopy. (B) Increased neural activity during TBI is represented by increased red:green emission (red pseudocolored to magenta) in the hindbrain of larvae (B″), compared to larvae not receiving TBI (B′) or fish not exposed to photoconverting light ('no PC' in panel B). These representative maximum intensity projection images show dorsal view of zebrafish brain (anterior at top), including merged, or red or green channels alone. (C) Heatmaps encode the CaMPARI signal (higher neural activity = higher red:green = hotter colors), highlighting location of increased neural activity during TBI relative to control larvae not receiving TBI. (D) Quantification of CaMPARI output in the hindbrain area reveals a significant increase in the neuronal excitability during TBI compared to control group not receiving TBI (**p=0.0087, Mann-Whitney test. Each data point is an individual larva). Scale bars = 100 μm.

aggregates that formed on the spinal cord as their abundance could be most efficiently quantified compared to aggregates that formed in the brain.

## Post-traumatic seizure intensity influences tauopathy progression

Considering the clinical prominence of post-traumatic seizures following TBI, and the suggested role of cell stress and increased neural activity in promoting protein misfolding diseases (*Kovacs et al., 2014*; *Sánchez et al., 2018*), we speculated that post-traumatic seizures might form a causal link between TBI and subsequent tauopathy. We first asked if a correlation exists between seizure intensity and extent of tauopathy. Following TBI, some larvae exhibited seizure-like movements, while some did not seem to move abnormally relative to untreated fish (*Figure 3H*). We sorted the larvae subjected to TBI into groups exhibiting the seizure-like behavior and those that displayed no overtly abnormal movement. Larvae exhibiting seizure-like behavior after TBI went on to develop abundant spinal cord aggregates (fivefold increase, p<0.001) in comparison to larvae that showed no seizure-like response to TBI (*Figure 5E*).

To assess the hypothesis that seizure activity has a causal role in increasing the abundance of Tau aggregates in our TBI model, we employed convulsant and anti-convulsant drugs to modulate the seizure intensity and in vivo neural activity. We selected drugs that are well-established to behave similarly in zebrafish as in mammals, although it is perhaps notable that the multi-day drug application used here is longer than the acute applications typically considered in zebrafish (*Ellis et al., 2012*). Our hypothesis predicted that decreasing seizure-like activity following TBI would reduce tauopathy. Indeed, applying the anti-convulsant drug Retigabine (RTG), that opens voltage-gated potassium channels (KCNQ, Kv7), resulted in a significant decrease in the abundance of GFP+ puncta (p=0.0107) with many TBI larvae not developing any Tau4R-GFP aggregates (*Figure 5F*).

Similarly, intensifying post-traumatic seizures via application of the convulsant kainate increased the abundance on tauopathy fourfold (p<0.0001. *Figure 5G*) in a dose-dependent manner (*Figure 5—figure supplement 5*). Kainate did not increase Tau4R-GFP+ puncta in the absence of TBI. Surprisingly, the convulsant 4-aminopyridine did not increase tauopathy (explored below).

To assess if the impacts of kainate and retigabine on tauopathy were directly due to their modulation of post-traumatic seizures, we applied effective doses of each in concert. Co-application of kainate and retigabine following TBI produced an abundance of Tau4R-GFP+ puncta that was indistinguishable from larvae receiving TBI without pharmacology (*Figure 5G*).

TBI-induced cell death was likewise correlated with the intensity of post-traumatic seizures. Co-application of kainate and retigabine following TBI increased or decreased, respectively, the abundance of cell death in a manner coordinate with the tauopathy (*Figure 5H*).

Overall, convulsant and anti-convulsant drugs acted to increase and decrease TBI-induced tauopathy, respectively. The drugs appear to be specific – their individual impacts on tauopathy and cell death are largely attributable to their epileptic and anti-epileptic modulation of post-traumatic seizures, because when kainate and retigabine were applied concurrently they negated each others' effects.

## Appearance of tauopathy following TBI requires endocytosis

To further examine increased seizure activity after TBI, we applied 4-aminopyridine (4-AP), a $K_v$ channel blocker and convulsant drug. We predicted that raising the level of seizure activity would elevate tauopathy abundance in our TBI model, aligning with our observations following application of kainate (above). Surprisingly, higher doses of 4-AP consistently abrogated the appearance of Tau aggregates. Treating TBI larvae with 200 or 800 μM of 4-AP for a prolonged period (38 hr,

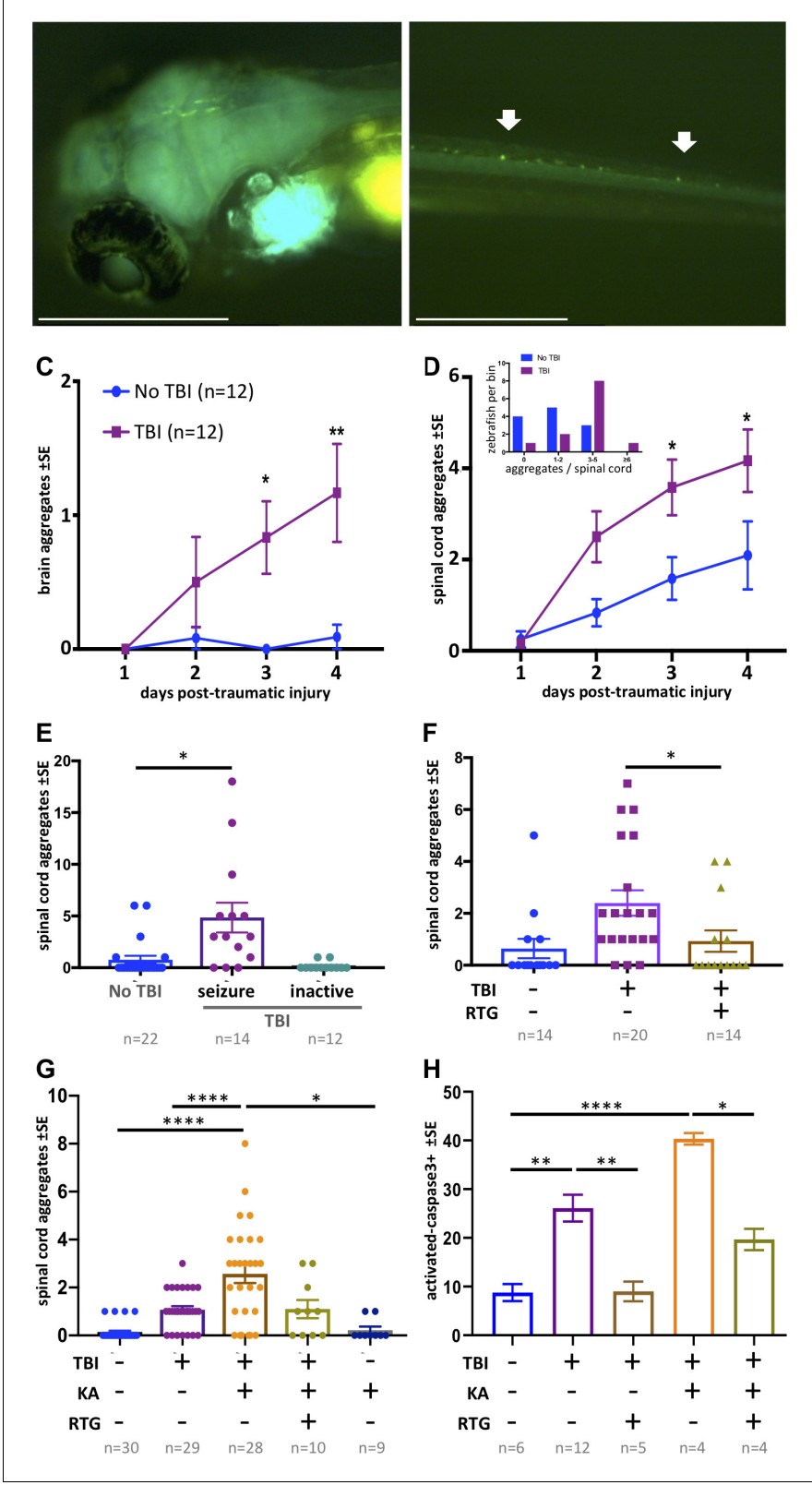

**Figure 5.** Traumatic brain injury (TBI) induces tauopathy in larval zebrafish. (**A**) GFP+ Tau puncta are detected in the brain of Tau4R-GFP biosensor zebrafish at 5 days post-traumatic brain injury (dpti). A 300 g weight was used to induce TBI throughout this figure. (**B**) Tau aggregates formed on the spinal cord as a result of the TBI as shown by arrows. (**C**) Tauopathy significantly increases over time following TBI compared to control group (No TBI)

*Figure 5 continued on next page*

*Figure 5 continued*

*p=0.0264, **p=0.007, two-way ANOVA with Tukey's multiple comparison test. (D) The number of Tau aggregates in spinal cord significantly increases over time following TBI compared to control group (*p=0.0224 at 3dpti and *p=0.0312 at 4dpti). Inset: Tau4R-GFP zebrafish larvae subjected to TBI develop more GFP+ puncta compared to the control group by 3dpti (inset plot is similar to *Figure 2E*). *Figure 5—figure supplements 2* and *3* plot this in individual fish. (E–H) Post-traumatic seizures link TBI to tauopathy. (E) Following TBI, larvae displaying post-traumatic seizures developed many more Tau aggregates relative to those not displaying post-traumatic seizures (**p=0.0011, Kruskal-Wallis ANOVA with Dunn's multiple comparison test). (F) Inhibiting post-traumatic seizures with the anti-convulsant retigabine (RTG, 10 µM) significantly decreased the abundance of GFP+ puncta in the spinal cord (p=0.0107, Mann-Whitney test). (G) Increasing post-traumatic seizure using the convulsant kainate (KA, 100 µM; see dose-response in *Figure 5—figure supplement 5*) significantly increased the formation of Tau aggregation following TBI; this effect was prevented by co-treatment with anti-convulsant RTG. ****p<0.0001, ordinary one-way ANOVA with Tukey's multiple comparison test. (H) Blunting post-traumatic seizures with RTG reduced TBI-related cell death. The main impact of RTG was specific to its anticonvulsant modulation of seizures because its effects were reversed by convulsant KA. Color scheme in panel C applies to other panels. n = number of zebrafish larvae. ****p<0.0001, ordinary one-way ANOVA with Tukey's multiple comparison test. Scale bars = 500 µm.

The online version of this article includes the following figure supplement(s) for figure 5:

**Figure supplement 1.** Traumatic brain injury (TBI) did not induce GFP+ puncta in transgenic zebrafish larvae expressing SOD1-GFP.

**Figure supplement 2.** Longitudinal analysis of individual fish after traumatic brain injury (TBI) shows various patterns of TAu inclusion formation and clearance in their brains.

**Figure supplement 3.** Longitudinal analysis of individual fish after traumatic brain injury (TBI) shows various patterns of Tau inclusion formation and clearance in their spinal cords.

**Figure supplement 4.** Increasing intensity of traumatic brain injury (TBI) significantly increased Tau4R-GFP puncta abundance, but only modest insignificant increases in Tau puncta were observed with an increasing number of successive brain injuries.

**Figure supplement 5.** Intensifying seizures following traumatic brain injury (TBI) increased abundance of GFP+ Tau puncta in a dose-dependent manner.

---

beginning 24 hr post-traumatic injury) significantly inhibited the abundance of Tau4R-GFP+ puncta in the TBI group (*Figure 6A–B* and its *Figure 6—figure supplement 1A-B*). Analysis of the distribution of larvae linked to the number of Tau aggregates supported this finding with no zebrafish larvae developing aggregates in groups treated with 4-AP (*Figure 6—figure supplement 1C*). It is worth noting that 4-AP is commonly used in zebrafish models of epilepsy, but rarely used for prolonged treatment. To evaluate if the time at which treatments are administered plays a role in this unexpected result, we treated larvae with 200 µM 4-AP at earlier time points, specifically during TBI and 1.5 hr later. We kept the duration of 4-AP treatment the same as previous experiments (38 hr). We found that administering 4-AP during different time windows relative to the TBI did not measurably alter the inhibitory action of 4-AP on the abundance of Tau aggregates (*Figure 6—figure supplement 1D*). A similar observation was made when the duration of the 4-AP treatment was reduced to 24 hr (*Figure 6—figure supplement 1E*).

Next, we considered if this unexpected inhibition of tauopathy by high-dose 4-AP convulsant is a direct consequence of increased neural activity (e.g. perhaps via neural exhaustion). We found that larvae receiving TBI and 4-AP continued to exhibit a lack of Tau aggregates when co-treated with anti-convulsant retigabine (p<0.0001) (*Figure 6B*). This suggested that high doses of 4-AP block the formation of Tau aggregates via a mechanism independent of its convulsant activity.

To resolve a mechanism whereby high doses of 4-AP reduced Tau pathology, contrary to our predictions above regarding neural hyperactivity, we considered previous in vitro work that demonstrated high concentrations of 4-AP cause reduced endocytosis of synaptic vesicles (*Cousin and Robinson, 2000*). To examine if the inhibitory actions of 4-AP on the abundance of Tau aggregates in our TBI model is consistent with an endocytosis inhibition mechanism, we treated our tau biosensor larvae post-traumatic injury with Pyrimidyn-7 (P7), a potent dynamin inhibitor that is known to block endocytosis (*McGeachie et al., 2013*), and analyzed the propagation of Tau pathology by quantifying the number of Tau inclusions. Owing to the potency of P7 and its impact on the survival of larvae, we treated the larvae with it for 24 hr at 3 µM. Similar to the findings with 4-AP, P7

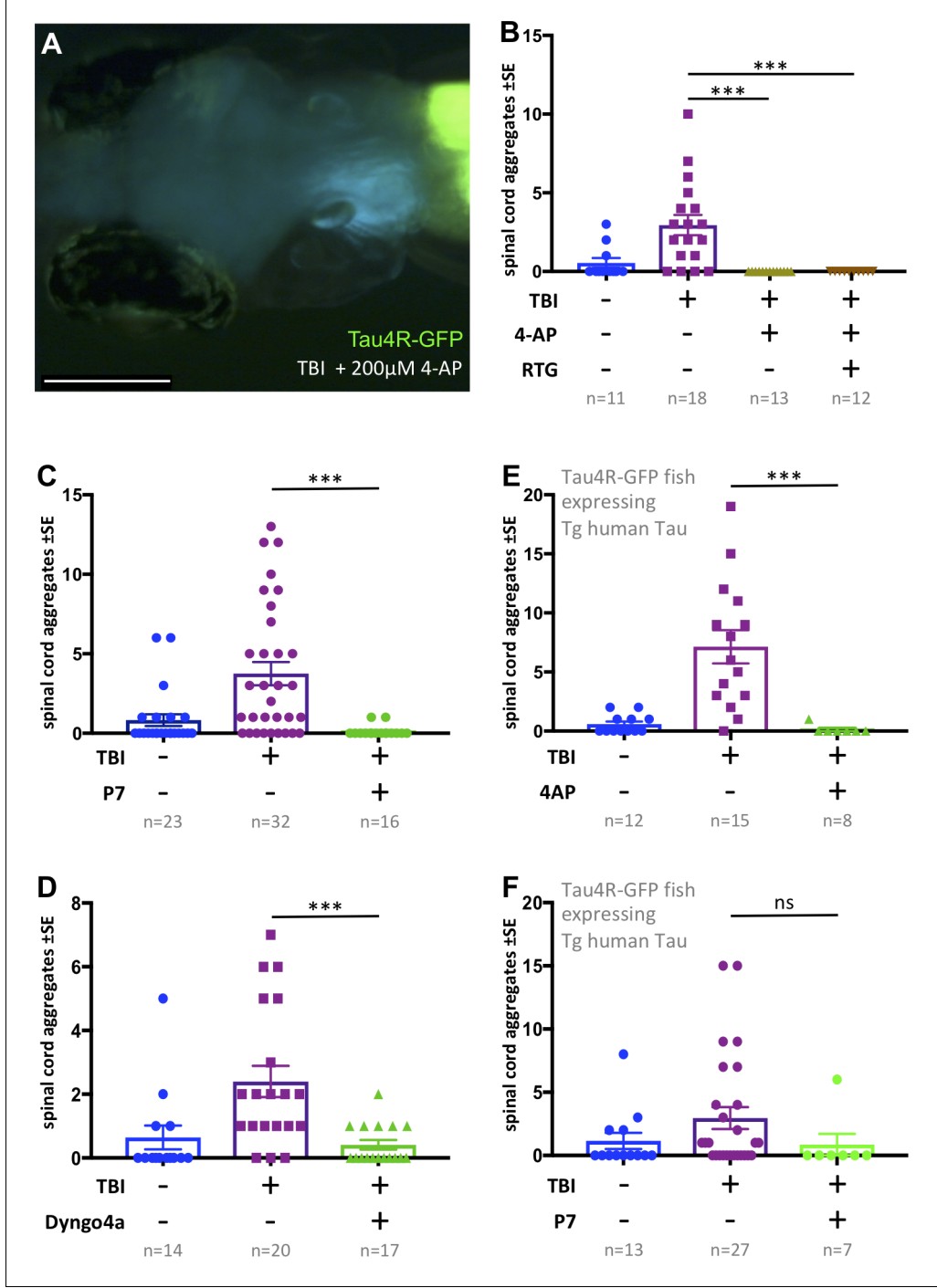

**Figure 6.** Tauopathy induced by traumatic brain injury (TBI) was attenuated by 4-aminopyridine (4-AP) via mechanisms independent of seizures. (**A**) Tau4R-GFP biosensor zebrafish larvae subjected to TBI and treated with the convulsant 4-AP show no brain puncta. Scale bar = 200 μm. (**B**) 4-AP significantly reduced (apparently eliminated) the abundance of GFP+ puncta in the brain and spinal cord compared to untreated TBI control. Results from alternative doses and timings of 4-AP are reported in *Figure 6—figure supplement 1*. The impact of 4-AP on tauopathy appears to be independent of its actions on post-traumatic seizures because reducing the latter with anti-convulsant retigabine (RTG) had no measurable effect. (**C-F**) Pharmacological inhibition of endocytosis reduced tauopathy following TBI. (**C**) Blocking endocytosis with Pyrimidyn-7 (P7) treatments significantly inhibited the formation of Tau4R-GFP+ puncta following TBI in zebrafish larvae (***p=0.001). (**D**) Dyngo 4a treatment significantly reduced Tau aggregates in the spinal cord (**p=0.0025) in a manner similar to P7. *Figure 6 continued on next page*

*Figure 6 continued*

(E) 4-AP treatment significantly inhibited the formation of Tau4R-GFP+ puncta in the spinal cord (***p=0.0003) of Tau biosensor line that also express human Tau (0N4R) after traumatic brain injury compared to untreated TBI control group. (F) A notable reduction in Tau aggregates was observed in the same line after treatment with P7 drug. Statistical analysis shows no significance difference (ns, p=0.2223) between groups. n = number of larvae. ****p<0.0001, Kruskal-Wallis ANOVA with Dunn's multiple comparison test used throughout this Figure.

The online version of this article includes the following figure supplement(s) for figure 6:

**Figure supplement 1.** 4-Aminopuridine (4-AP) abrogates TBI-induced Tau aggregates when applied at various doses or for various times.

treatments significantly inhibited the formation of Tau4R-GFP+ puncta in TBI larvae (p=0.001) (*Figure 6C*). We assessed further the role of endocytosis by employing another dynamin- inhibitor drug, Dyngo 4a, that is less potent than P7 (*McCluskey et al., 2013*). We obtained similar results in which Dyngo 4a treatments significantly reduced tauopathy in our TBI model (*Figure 6D*). To determine if these results are applicable to human tau, we induced traumatic brain injury on double-transgenic larvae expressing both human Tau (the 0N4R human Tau isoform, see *Bai et al., 2007*) and our Tau biosensor reporter, followed by treatment with either 4-AP or P7. Apart from the untreated control, both groups treated with 4-AP or P7 exhibited a noticeable reduction in abundance of Tau aggregates. While the decrease in the case of P7 was not statistically significant, statistical analysis showed significance after 4-AP treatments (p=0.0003) (*Figure 6E and F*). These findings confirmed the ability of 4-AP and dynamin inhibitors of reducing human Tau aggregates in our TBI larvae.

## Discussion

The consequences of concussive blasts and TBI extend beyond the proximate injury – they are prominent risk factors for devastating dementias including AD, CTE and other tauopathies. Identifying the causal links that entwine TBI with subsequent tauopathies would inspire improved diagnostics and therapeutics. Investigating these mechanisms has been hampered by lack of tractable models, since cell culture platforms cannot faithfully represent the injury or the response-to-injury or the treatment thereof. Indeed, TBI and tauopathies are complex tissue and systems-level events with pathobiology progressing on a backdrop of dynamic vigorous neural function, prion-like vectoring of misfolded proteins via glymphatic and blood vasculature, immune and support cells, sleep physiology, homeostatic regulation and complex drug metabolism. Rats have been the favored animal model for TBI, and mice can complement this as insightful models of tauopathy, yet both are challenged by expense, ethical considerations, and CNS tissues that are relatively inaccessible to (longitudinal) visualization of cellular events in living individuals (*Bodnar et al., 2019*; *Marklund, 2016*; *Meconi et al., 2018*; *Pham et al., 2019*). Here, our zebrafish models replicate human TBI and tauopathy; the model is imperfect (e.g. see *Limitations* below) but by addressing many of these challenges in an accessible and vibrantly active vertebrate brain, we offer an innovative approach for the study of prion-like events, tauopathy and/or TBI.

Here, we introduce a simple method for delivering TBI to larval zebrafish that can be scaled to high-throughput and adopted at low expense. The tractability and transparency of zebrafish larvae allowed us to deploy genetically encoded fluorescent reporters that were validated to (i) uniquely quantify neural activity on freely behaving animals *during* TBI and (ii) effectively document prion-like tauopathy in individual subjects over multiple days. The accessibility of this platform to pharmacology allowed us to query cell biology events in vivo and support a role for endocytosis in prion-like progression and TBI-induced tauopathy. Further, anti-convulsant drugs were potent mitigators of the tauopathy and cell death that emerged subsequent to TBI (schematized in *Figure 7*); these effects were attributable to suppression of post-traumatic seizures (as proven by the anticonvulsant's therapeutic effects being reversed by co-application of convulsant drugs). It remains to be seen if these data have any bearing on the long-term clinical management of TBI patients, and we newly speculate that prophylactic application of anti-convulsants (already common for blunting of patient's post-traumatic seizures) might hinder progression of tauopathies including CTE and AD. If true, then debates regarding the optimal regimen of anti-epileptics for TBI patients should consider their potential for providing long-term benefits on dementias.

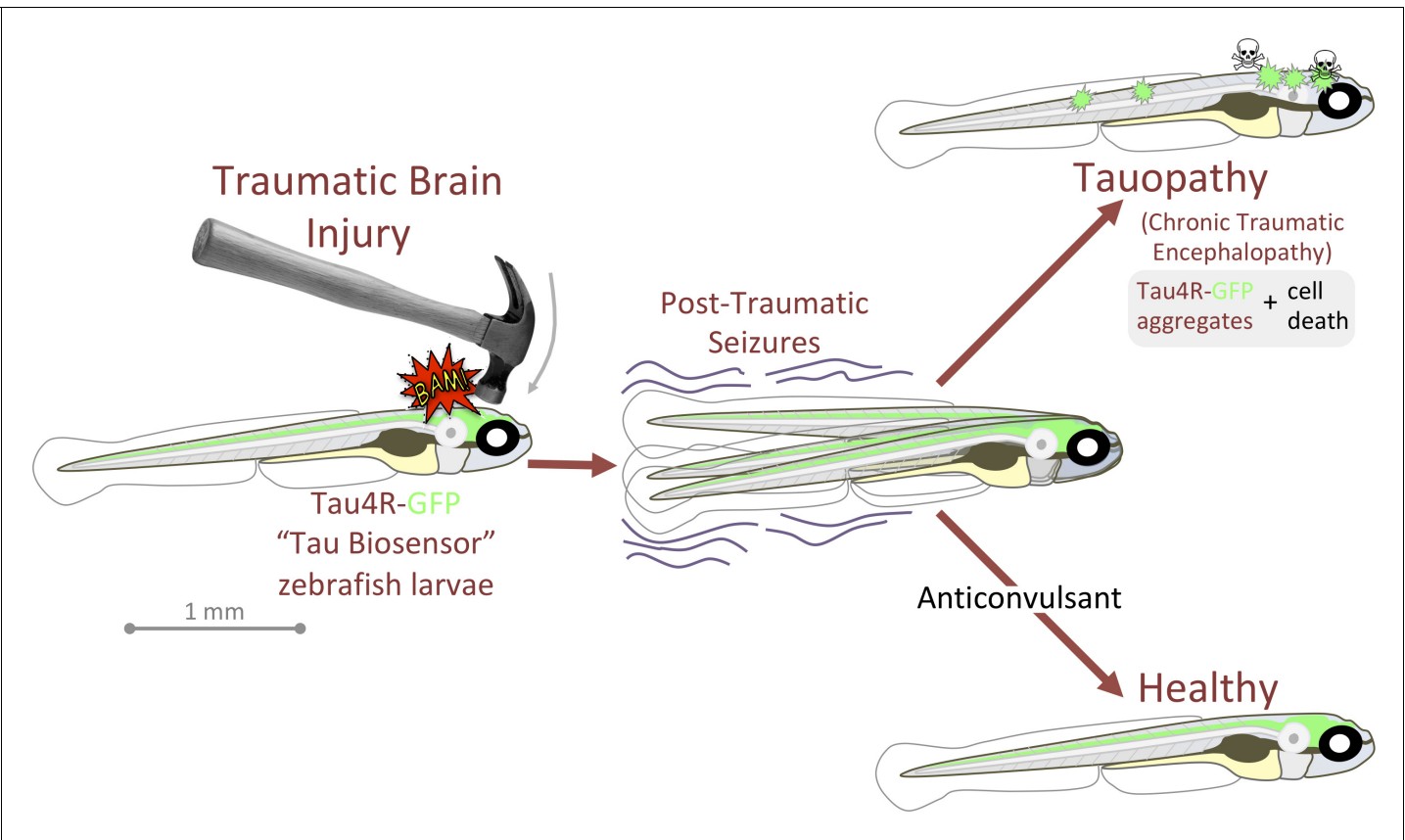

**Figure 7.** Graphical summary: anticonvulsants reverse the tauopathy and cell death exhibited by zebrafish larvae following traumatic brain injury (TBI). Tauopathy was reported via aggregation of a genetically encoded chimeric protein, Tau4R-GFP, that was expressed throughout the central nervous system. TBI led to seizures, and subsequently to Tau aggregation and cell death (akin to chronic traumatic encephalopathy, CTE). The tauopathy and cell death were ameliorated, producing healthy larvae, by blocking seizures with anti-convulsants; these effects were specific insomuch that they could be reversed by co-application of convulsants.

### Prion-like tauopathy induced by TBI

It is well established that TBI induces tauopathy, and over the past decade, numerous in vitro and in vivo studies have supported that Tau proteins possess prion-like properties. Indeed seeding, templated misfolding (conversion) and spread to synaptically connected regions has been documented in tauopathies such as AD and FTD (*Ayers et al., 2018*; *de Calignon et al., 2012*; *Goedert et al., 2017a*; *Goedert et al., 2017b*; *Iba et al., 2015*; *Woerman et al., 2016*). Various mechanisms have been proposed for the transcellular transfer of Tau seeds including release mechanisms via exosomes, or cellular uptake mechanisms via endocytosis (*Demaegd et al., 2018*; *Evans et al., 2018*; *Wang et al., 2017*; *Wu et al., 2013*). Nonetheless, these suggested mechanisms were postulated based on in vitro evidence as there is a lack of appropriate models that can visualize and manipulate the prion-like spread of Tau pathology between tissues in a vibrant brain. It is unclear if these mechanisms are universal in progression of all tauopathies or if there are factors and mechanisms that are unique to each disease. Our zebrafish models allow us to study the progression and spread of TBI-induced tauopathy longitudinally in living animals, an experimental advantage that is unmatched among TBI animal models.

Regarding TBI, a large knowledge gap exists regarding how Tau seeds are released and/or internalized by adjacent (or far-flung) cells - indeed the prion-like properties of Tau species following TBI had not been assessed until very recently (*Woerman et al., 2016*; *Zanier et al., 2018*). Moreover, the focus in the literature has mostly been directed toward repetitive mild trauma as it is most associated with CTE, yet the various forms of TBI all are considered risk factors for neurodegeneration. The recent revelation that most TBI patients, whether they suffered from single or repetitive brain

trauma, all exhibited Tau pathology similar to CTE (*Washington et al., 2016*; *Zanier et al., 2018*) suggests all forms of TBI might incorporate tauopathies. Our mode of TBI on larval zebrafish entails a pressure wave that most closely mimics a blast injury (e.g. as experienced by military personnel or civilians near an explosion, and presumably producing injury throughout the body), but the etiology leading to tauopathy probably has many similarities regardless of the mode of the initiating TBI.

## Post-traumatic seizures accelerate tauopathy following TBI

We inspected the role of seizure activity and/or neuronal excitability, as well as the role of dynamin-dependent endocytosis, during the progression of tauopathy after TBI. We focussed our attention on seizure activity in part because seizures frequently occur in TBI patients following blast traumatic injury (*Englander et al., 2014*; *Kovacs et al., 2014*). We hypothesized that neuronal excitability and seizure activity after TBI can play a role in accelerating the wide dissemination of Tau pathology. As such, we introduced two new approaches to test this hypothesis. The first approach was to engineer a novel in vivo Tau biosensor model in zebrafish that can visualize pathological Tau spreading and accumulation within the intact and vibrant CNS. The Tau biosensor zebrafish express human tau4R-GFP reporter protein, and we confirmed its ability to detect Tau seeds from various sources both in vivo and in vitro, similar to previously engineered in vitro models (*Kaufman et al., 2016*; *Sanders et al., 2014*). Our second approach was to introduce and optimize an elegantly simple technique to cause pressure-wave induced TBI, similar to human blast TBI, in zebrafish larvae. We endeavoured to inflict injury on larval zebrafish rather than adults because of the synergistic advantages that larval zebrafish provide: these include economical access to large numbers of individuals and associated statistical power, and the tractability of larvae for high-throughput in vivo screening of therapeutic agents (*Saleem and Kannan, 2018*). Larval zebrafish provide a large economic advantage compared to adults, with respect to time, cost per individual and space consumed in animal housing. Moreover, injuring animals in larval stages is viewed as an ethically favorable Replacement [sensu 'the three Rs' of *Russell and Burch, 1959*] compared to injuring adult subjects. Thus, regardless of any bioethical considerations based on taxonomy, larval fish (that are accessible early in their development via external fertilization of eggs) are ethically advantageous to rodents (that are accessible for TBI only at postnatal stages) when considering highly invasive procedures like TBI. The latter conclusion relies on the assumption that the knowledge gained is of value, that is relevant to appreciating disease etiology.

Our data argue that our TBI methods are germane to clinical aetiology, because (akin to existing animal models of TBI) we were able to confirm the presence of various markers associated with brain injury, such as cell death, abnormalities in blood flow, hemorrhage and the occurrence of post-traumatic seizures. Recently, post-traumatic seizures were also observed in *adult* zebrafish when TBI was delivered via focused ultrasound (*Cho et al., 2020*) offering an additional opportunity for zebrafish to be an important tool and bridge organism in the field of neurotrauma.

The post-traumatic seizures apparent in our TBI model led us to consider the neural events occurring during the TBI, and their potential bearing on the correlation between neural activity and tauopathy. Few studies examine how TBI impacts neuronal circuits, especially in vivo, and these typically consider events several hours or days after brain trauma (*Bugay et al., 2020*). This may be of importance when considering evaluating the reasons behind the developments of post-traumatic seizures and epilepsy. In a controlled cortical impact model of TBI, an initial decrease or loss in neuronal activity is recorded after injury before a rise in neuronal activity is noted (*Ping and Jin, 2016*). Whether this occurs in different types of TBI, like blast TBI, was unexamined. To address this, we performed TBI on larval zebrafish expressing CaMPARI, a genetically encoded optogenetic reporter of neural activity. CaMPARI is particularly ideal for this question, as its reportage of neural activity (a stable and quantifiable shift from green to red fluorescence) occurs only during user-defined times and that reportage is relatively permanent. This allowed us to quantify the CNS activity that had occurred during TBI, by characterizing the ratio of red:green fluorescent emission using confocal microscopy after the TBI injury was completed. This approach therefor allows relatively easy access to quantifying neural activity *during* injury in an unencumbered freely swimming animal. Here, we revealed for the first time a snapshot of neurons becoming active at the moment of TBI. Our results demonstrated an increase in neuronal excitability upon TBI, which may contribute to the frequency of post-traumatic seizures observed in our model, other blast TBI models and TBI patients (*Bugay et al., 2020*). The increases in neuronal activity were especially prominent in the hindbrain;

this may be due to this region being susceptible to injury in our paradigm, and/or represent an output that is related to increased movement (e.g. this region is home to robust motoneurons that would be hyperactive during seizures). It will be of interest to resolve what types of neurons and neurochemistry are most impacted by the injury. Notably, the subsequent Tau4R-GFP aggregates were also most prominent in this hindbrain region, and while this could be coincidental, or perhaps an artefact of a hidden differential Tau4R-GFP abundance, it is also possibly a result of increased neural activity. Regarding the etiology of tauopathy subsequent to TBI, the CaMPARI quantification provided us important validation that neural activity was substantively impacted by TBI, complementing the evidence of increased seizure-like movements. This supported our rationale that convulsant and anti-convulsant drug treatments might modulate neural activity and thereby accelerate or decelerate tauopathy accumulation.

Indeed, we had noted the occurrence of post-traumatic seizures in most of our TBI samples, which is in agreement with the prevalence of seizures in blast TBI patients and TBI rodent models (*Bugay et al., 2020*; *Kovacs et al., 2014*). However, whether post-traumatic seizures contribute to prion-like spreading of Tau pathology (observed after TBI or not) was unknown. Beyond TBI, several investigations have supported an association between Tau pathology and seizures (*Sánchez et al., 2018*; *Tai et al., 2016*). Studies on epileptic human temporal structure revealed accumulation of Tau aggregates (*Sánchez et al., 2018*). In 3XTg AD mice, induced chronic epilepsy was associated with changes of inter-neuronal p-Tau expression (*Yan et al., 2012*). Additionally, data obtained from postmortem analysis of patient tissues with AD and drug resistant epilepsy uncovered a correlation between symptomatic seizures, increased Braak staging and accelerated Tau accumulation (*Thom et al., 2011*). Interestingly, the presence of tau deposits in epileptic patients and the similarity of its pathology to CTE suggest a conceivable role for seizures influencing the progression of Tau pathology in a similar manner to TBI (*Puvenna et al., 2016*). Indeed, our data from the application of the convulsant kainate here support the role of post-traumatic seizure in enhancing Tau abundance and cell death in our TBI model (*Figure 5G,H*). This finding is in line with observations in a patient with epilepsy and a history of head injury, in which progressive Tau pathology was noted (*Geddes et al., 1999*; *Thom et al., 2011*). Intriguingly, reducing seizure activity after TBI via anti-convulsant drugs was able to significantly reduce tauopathy and cell death, providing further evidence of the relationship between seizures and tauopathy in TBI (*Figure 5F,G and H*). The mechanism of drug action appears to be dominated by its anticonvulsant properties, because its effects were reversed by co-application of convulsants. Thus, anti-convulsants are intriguing as a route to slowing progression of tauopathy following TBI, and it is encouraging that they are already commonly deployed to prevent post-traumatic seizures.

## Endocytosis mediates prion-like spread of tauopathy

One particular convulsant drug, 4-AP, inhibited tauopathy in our TBI model (*Figure 6A and B*), contrary to our hypothesis that seizure intensity is positively correlated with tauopathy following TBI. 4-AP is a voltage-gated potassium channel blocker that enhances neuronal firing activity and has been used often in zebrafish seizure studies (*Kanyo et al., 2020b*; *Kasatkina, 2016*; *Liu and Baraban, 2019*; *Lundh, 1978*; *Winter et al., 2017*). Yet, 4-AP is rarely administered for prolonged treatments such as those we deployed here, for example past studies rarely exceed one hour of 4-AP (*Winter et al., 2017*). Thus, we considered that our high dose and prolonged stimulation with 4-AP may have led to off-target effects; we confirmed this insomuch that the 4-AP's inhibition of tauopathy was not related to its convulsant properties (as determined by 4-AP's effects being unaltered by potent anti-convulsants (*Figure 6B*)). Indeed, previous in vitro work revealed high concentrations or prolonged stimulation with 4-AP has off-target effects via inhibiting dynamin, which is important for the endocytosis of synaptic vesicles at the nerve terminals (*Cousin and Robinson, 2000*). The inhibition of endocytosis observed in that study was independent of 4-AP-dependent seizure activity.

We further queried the potential role of dynamin-dependent endocytosis in the prion-like progression of Tau pathology after TBI by applying endocytosis inhibitors that target dynamin. Dynamin is a GTPase involved in two mechanisms of endocytosis that are important for synaptic vesicle transport (*Singh et al., 2017*). Empirical work on human stem-cell-derived neurons has indicated that Tau aggregates are internalized via dynamin-dependent endocytosis and that blocking other endocytosis pathways independent of dynamin, such as bulk endocytosis and macropinocytosis, did not disrupt

Tau uptake (*Evans et al., 2018*). On the contrary, inhibiting dynamin significantly decreased the internalization of Tau aggregates. Our results are in line with the previously mentioned findings that show Tau progression in TBI models depends on dynamin-dependent endocytic pathways - blocking them with two different inhibitors (and with 4-AP) dramatically lessened the abundance of Tau seeds (*Figure 6C–F*). Hence, our findings not only provide in vivo validation of past in vitro works, but also suggest mechanisms underlying prion-like spreading of Tau seeds in TBI and CTE that could aid in developing therapeutic strategies.

## Limitations of our approach

Our tauopathy biosensor, human Tau4R-GFP, was deployed in vivo and uniquely able to detect significant increases (and decreases) in the abundance of Tau aggregates following various insults and treatments, typically in a dose-dependent manner and in harmony with expected trends. Considering this success, it remains perplexing and intriguing that a subset of larvae exhibit GFP+ Tau puncta despite receiving no known tauopathy-inducing insults. This suggests that the larvae express the transgene at a level near to a threshold for producing spontaneous aggregates. We performed selective breeding to minimize these occurrences and tentatively believe, after too few generations, that genetics of the fish is a factor – substantial genetic variation exists in zebrafish in-bred lines (*Balik-Meisner et al., 2018*; *Guryev et al., 2006*). However, we acknowledge the variation could be a minor technical artefact rather than biological. It remains to be determined if our biosensor can self-aggregate in some situations, or if these aggregates represent transient moments of imbalance in the proteostatic cycle of Tau aggregation and clearance. Regardless of this variation amongst individual larvae, the effects of neurotrauma and drug interventions were rational and robust when considering the mean of several animals; the large sample sizes available with the larval zebrafish model can overcome some of this variability. More optimistically, this inter-individual variation and stochastic appearance of tauopathy, in a high-throughput model, could be leveraged to newly appreciate aspects of spontaneous AD or other non-familial tauopathies. Regardless, future work will also need to characterize the biochemistry and biophysics of the human Tau inclusions in zebrafish compared to patients or rodent models.

Regarding our TBI methods, further refinements may yet be able to improve consistency of the injury and reduce the apparent variability between individuals. This variability is real, but somewhat offset by the large sample sizes attainable: our TBI methods offer the potent advantages of zebrafish larvae with respect to genetic and drug accessibility in high-throughput formats, while also retaining the critical in vivo complexity required to investigate disease etiology and treatments. Further, it remains to be established if the mechanisms we reveal are ubiquitous across the various forms of TBI: our model fills a gap by supplying a rare 'closed head' TBI model (as opposed to the majority of animal models that access the brain by removing skull elements prior to brain injury, see exceptions by *Meconi et al., 2018*; *Mychasiuk et al., 2014*). Our model might be most relevant to brain trauma experienced by the human fetus (e.g. during car collisions or domestic abuse), considering the developmental stage and aqueous media. Moreover, the injury induced herein is presumably not limited to the brain, and probably injures various tissues including the spinal cord; this should be considered when interpreting the data. Further work is also needed to appreciate how the physics of our blast injury is altered by occurring at a small scale (e.g. larval brain is <500 µm). At this point, we are left to assume that the cellular and physiological aspects of TBI we consider here are sufficiently similar across all classes of TBI, and thus the knowledge gleaned may be variably applicable.

Finally, we have chosen to restrict our analysis to study of larval fish. While this offers many logistical and ethical advantages detailed above, it limits our study to acute effects occurring over the course of several days. Conclusions from such work, once refined and validated using the power of the in vivo zebrafish larva model, should be tested in rodent models where it is equally time-consuming to assess the long-term efficacy of treatments on these progressive late-onset dementias.

## Conclusion

Currently, no available treatments are applicable to all tauopathies, which remain as devastating and inevitably fatal dementias. Zebrafish larvae, fostered by appropriate innovations, now offer a potent complement both to rodent models of TBI and to cellular models of tauopathy. Our engineered fish allowed us to reveal post-traumatic seizures as a druggable mechanistic link between TBI and the

prion-like progression of tauopathy. Intriguingly, our conclusions have potential for translation to TBI clinics where anti-convulsants are already in use as prophylactics for post-traumatic epilepsy, though further work remains to address if they mitigate (the risk or severity of) later progression of CTE, AD or other tauopathies.

# Materials and methods

### Key resources table

| Reagent type (species) or resource | Designation | Source or reference | Identifiers | Additional information |
|---|---|---|---|---|
| Strain (zebrafish) | Tg(eno2:hsa.MAPT-ires-egfp)$^{Pt406}$ | Burton's Lab (*Bai et al., 2007*) | ZFIN ID: ZDB-ALT-080122–6 | Zebrafish that express human four repeat TAU |
| Strain (zebrafish) | Tg(eno2:SOD1-GFP)$^{ua3181}$ | This paper | N/A | zebrafish biosensor engineered to detect human SOD1 aggregation |
| Strain (zebrafish) | Tg[elavl3:CaMPARI(W391F+V398L)]$^{ua1344}$ | In house allele (*Kanyo et al., 2020b*) established using vector provided by Eric Schreiter's lab | | zebrafish expressing the calcium sensor CaMPARI |
| Strain (Zebrafish) | Tg(eno2:Hsa.MAPT_Q244-E372−EGFP)$^{ua3171}$ | This paper | N/A | Zebrafish biosensor engineered to detect human Tau aggregation |
| Genetic reagents (zebrafish) | Multisite Gateway technology (BP Clonase II Enzyme mix and LR Clonase II Plus enzyme) | Thermo Fisher | Cat# 11789020 Cat# 12538120 ZFIN ID: ZDB-PUB-170809–10 | *Guo and Lee, 2011*; *Kwan et al., 2007* |
| Cell line (*Homo-sapiens*) | HEK293T | ATCC Provided by Dr. David Westaway's laboratory | Cat# CRL-3216, RRID:CVCL_0063 | |
| Sequence-based reagent | GFP_R | This paper | PCR primer | TCTCGTTGG GGTCTTTGCTC |
| Biological sample (mouse) | Whole brains | Tissues were provided by Dr. David Westaway (*Eskandari-Sedighi et al., 2017*; *Murakami et al., 2006*) | | Isolated from wild -type mice with 129/SvEvTac genetic background, and TgTauP301L mice |
| Antibody | Anti GFP (rabbit monoclonal) | Abcam | Cat# ab183734, RRID:AB_2732027 | WB(1:3000) |
| Antibody | Anti-β-actin (rabbit polyclonal) | Sigma-Aldrich | Cat# A2066, RRID:AB_476693 | WB (1:10000) |
| Antibody | Anti-Active-Caspase-3 (rabbit polyclonal) | BD Pharmingen | Cat# 559565, RRID:AB_397274 | IHC (1:500) |
| Antibody | Alexa Fluor 647 (chicken anti-rat IgG) | Invitrogen | Cat# A-21472, RRID:AB_2535875 | WB (1:500) |

*Continued on next page*

*Continued*

| Reagent type (species) or resource | Designation | Source or reference | Identifiers | Additional information |
|---|---|---|---|---|
| Peptide, recombinant protein | Human MAPT (2N4R) | rPeptide | Cat# T-1001–2 | Resuspended to 2 mg/ml before use |
| Chemical compound, drug | Kanic acid monohydrate | Sigma Aldrich | K0250 | |
| Chemical compound, drug | Retigabine | Toronto research chemicals | R189050 | |
| Chemical compound, drug | 4-Aminopyridine (4AP) K+ channel blocker | Sigma | Cat# 275875–1G | |
| Chemical compound, drug | Pyrimidyn-7 (P7) Dynamin inhibitor | Abcam | Cat# ab144501 | 50 mM concentration supplied in DMSO |
| Chemical compound, drug | Dyngo 4a | Abcam | Cat# ab120689 | |
| Software, algorithm | Lab Chart 7 (software) | AD Instruments | | |
| Software, algorithm | Geneious Prime (bioinformatics software) | geneious.com | Version 8 | |
| Other | Human MAPT (Gene block) | Ordered from IDT | | This is aa 244–372 of the full-length TAU 2N4R with seven-amino acid C-terminal linker (RSIAGPA) |
| Other | Power lab Data acquisition device (equipment) | AD Instruments | 2/26 | |
| Other | Piezoresistive pressure transducer (equipment) | AD Instruments | Cat# MLT844 | |
| Other | Lipofectamine. 2000 (transfection reagents) | Invitrogen | Cat# 11668–019 | |
| Other | DAPI stain | Thermo Fisher | D1306 | (1 μg/mL) |

## Animal ethics and zebrafish husbandry

Zebrafish were raised and maintained following protocol AUP00000077 approved by the Animal Care and Use Committee: Biosciences at the University of Alberta, operating under the guidelines of the Canadian Council of Animal Care. The fish were raised and maintained within the University of Alberta fish facility under a 14/10 light/dark cycle at 28°C as previously described (*Westerfield, 2000*).

## Generating transgenic tauopathy reporter zebrafish

To engineer the transgenic Tau4R-GFP reporter zebrafish, the human wild-type MAPT sequence of the four-microtubule binding repeat domain (aa 244–372 of the full-length TAU 2N4R, NCBI NC_000017.11, protein id NP_005901) with a seven-amino acid C-terminal linker (RSIAGPA) was ordered as a gene block from IDT. The gene block was subcloned into a middle entry cloning vector (Multisite Gateway technology, ThermoFisher). This was recombined with the p5E-enolase2 and

p3E-GFP components into destination vector pDestTol2CG2 of the Tol2kit (*Guo and Lee, 2011*; *Kwan et al., 2007*). The destination vector contained a reporter construct [encompassing EGFP driven by the cardiac myosin light chain (clmc) promoter that helps identify stable transgenic zebrafish]. The resulting plasmid pDestTol2CG2.eno2:Tau4R-GFP was delivered in a 10 µl injection solution including 750 ng/µl of the construct mixed with 250 ng/µl Tol2 transposase mRNA, 1 µl of 0.1M KCL, and 20% phenol red. The solution was injected into the single-cell embryos of Casper zebrafish line (transparent zebrafish line)(*White et al., 2008*). Injected embryos were screened for mosaic expression of the Tau4R-GFP transgene at 2 days post-fertilization (dpf) using a Leica M165 FC dissecting microscope. F0 mosaic fish were raised to adulthood and outcrossed. Successful F1 embryos were identified by their abundant expression of Tau4R-GFP in the CNS and the green heart marker. The stable transgenic line *Tg(eno2:Hsa.MAPT_Q244-E372−EGFP)*[ua3171] was assigned the allele number ua3171.

An equivalent transgenic zebrafish biosensor was engineered to detect human SOD1 aggregation. Subcloning from existing vectors (*Pokrishevsky et al., 2018*) produced pDestTol2CG2.eno2:SOD1-GFP and similar transgenesis methods engineered the *Tg[eno2:SOD1-GFP]* zebrafish line that was assigned allele number ua3181.

## Cell culture and generation of tauopathy reporter stable cell line

To move the Tau4R-GFP reporter above into a vector appropriate for cell culture, BamHI and Xhol nuclease restriction enzymes were employed to remove the Tau4R-GFP fragment from pDest tol2CG2.eno2.Tau4R-GFP.pA. The Tau4R-GFP fragment was subcloned into the pCDNA3.1 vector using a T4 DNA ligase enzyme. Sequencing of the cloned vector with the following reverse primer for GFP (TCTCGTTGGGGTCTTTGCTC) confirmed the proper orientation. Purification of the plasmid was conducted with the Qiagen purification kit. HEK293T cells were grown in Dulbecco's modified Eagle's medium (GibcoTM, ThermoFisher) supplemented with 10% fetal bovine serum and 1% penicillin/streptomycin. All cells were maintained at 37°C in a humidified 5% $CO_2$ incubator. For passaging cells, cells were washed with phosphate-buffered saline (PBS) before trypsinization with 0.05 Trypsin-EDTA (Sigma Aldrich, T4174).

HEK293T cells were plated at $1 \times 10^6$ cells/well in six-well plates. Cells were transfected with pcDNA3.1.Tau4R-GFP plasmid 24 hr after plating using lipofectamine 2000 reagents according to the manufacturer's guidelines. Briefly, 4 µg of pcDNA3.1.Tau4R-GFP was diluted in 250 µl of Opti-MEM media (GibcoTM, ThermoFisher). The expression of the fluorescent reporter was confirmed the next day through microscopic analysis. A stable cell line was established by replating the transfected cells at a 1:10 dilution and selecting in DMEM media containing 1200 µg/ml geneticin (GibcoTM, ThermoFisher). Expression of the fused fluorescent proteins in the stable cell lines was confirmed using fluorescent microscopy. Polyclonal cells and monoclonal cells were grown to confluency in 10 cm dishes, then stored in liquid nitrogen until use.

## Immunoblotting of cell lysate and zebrafish brain lysate

For cell lysate preparation, cells were washed with cold PBS, then collected and incubated with cold lysis buffer (150 mM NaCl, 50 mM Tris-HCl (pH 8), 1 mM EDTA and 1% Nonidet P-40) supplemented with protease inhibitor (Cocktail Set III; Millipore) for 10 min on ice. Cells were lysed using a bio-vortexer homogenizer for 20 s for two rounds. The lysate was centrifuged at 13,000 rpm for 10 min at 4°C. The supernatant was collected, and the protein concentration was determined using the Qubit Protein Assay Kit (Invitrogen).

For zebrafish brain lysate preparation, the brains of adult zebrafish were dissected. Brains were homogenized in cell lysis buffer (20 mM HEPES, 0.2 mM EDTA, 10 mM NaCl, 1.5 mM $MgCl_2$, 20% glycerol, 0.1% Triton-X) with protease inhibitor and phospSTOP (Sigma-Aldrich) in the case of pt406 Tg. Brains were lysed using a bio-vortexer homogenizer and sonicated for 3 s for one round. Samples were centrifuged as above and concentration of the samples was assessed in a Qubit fluorometer (Invitrogen).

For immunoblotting, 30–40 µg of the total protein was combined with 2X sample buffer (Sigma-Aldrich) and boiled for 10 min before loading in 11% SDS-PAGE. Electrophoresis was performed using the Bio-Rad Power PAC system in running buffer (25 mM Tris base, 192 mM glycine and 0.1% SDS). The gel was transferred to a PVDF membrane using a wet transfer system. All membranes

were blocked for one hour in protein-free blocking buffer PBS (ThermoFisher) or TBST with 5% milk and then incubated with primary antibody overnight at 4°C with gentle agitation. The primary antibodies used in this study include rabbit monoclonal GFP (abcam, EPR14104) at 1:3000 dilution, rabbit anti-β-actin (Sigma-Aldrich, A2066) at 1:10,000. All membranes were washed three times with 1X TBST before incubation with secondary antibody (goat-anti-mouse) HRP or HRP-conjugated anti-rabbit at 1:5000 dilution (Jackson ImmunoResearch) for 1 hr at room temperature. The membranes were washed for the final time before visualization using Pierce ECL Western Blotting Substrate (ThermoFisher) on a ChemiDoc (Biorad). For stripping and re-probing, the membranes were stripped using mild stripping buffer (199.8 mM Glycine, 0.1% SDS, and 1% Tween 20 with a pH of 2.2) before blocking them and repeating the methods described before.

## Immunohistochemistry

Larvae were fixed overnight in 4% paraformaldehyde, either 1 day after being subjected to TBI or following the subsequent application of drugs as indicated. Immunostaining of Activated-Caspase3 on whole-mount larvae was carried out as previously described (*Duval et al., 2014*). Larvae were washed with 0.1 M PO$_4$ with 5% sucrose three times before washing with 1% Tween in H$_2$O (pH 7.4), and then −20°C acetone. Larvae were incubated in PBS3+ containing 10% normal goat serum for 1 hr and then incubated with primary antibody with 2% normal goat serum in PBS3+. The primary antibody used was polyclonal Anti-Active-Caspase-3 (BD Pharmingen, 559565) at 1:500 dilution. The secondary antibody applied was Alexafluor 647 anti-rabbit at 1:200 dilution (Invitrogen). Larvae were counterstained with 4′,6-diamidino-2-phenylindole (DAPI) (ThermoFisher) for 30 min.

## Preparations of mouse brain homogenate (crude and PTA precipitated)

Brains from TgTau$^{P301L}$ mice and non-Tg littermate controls (129/SvEvTac genetic background) were provided by Dr. David Westaway and Dr. Nathalie Daude (*Eskandari-Sedighi et al., 2017*; *Murakami et al., 2006*). Crude brain homogenate was prepared by homogenizing the brains to 10% (wt/vol) in calcium- and magnesium-free DPBS that included a protease inhibitor and phosSTOP, using a glass homogenizer and power gen homogenizer (Fisher Scientific). Samples were then centrifuged at 13,000 rpm for 15 min at 4°C. The clear supernatant was collected, aliquoted and stored in −80°C until use for experiments.

The phosphotungstate anion (PTA)-precipitated brain homogenate was prepared as described (*Woerman et al., 2016*). Briefly, 10% (wt/vol) brain homogenate was prepared as reported above and mixed with a final concentration of 2% sarkosyl (Sigma Aldrich) and 0.5% benzonase (Sigma Aldrich, E1014), and then incubated at 37°C for two hours with constant agitation in an orbital shaker. Sodium PTA (Sigma Aldrich) was dissolved in ddH$_2$O, and the pH was adjusted to 7.0 before it was added to the samples at a final concentration of 2% (vol/vol). The samples were then incubated overnight under the previous conditions. The next day, the samples were centrifuged at 16,000 g for 30 mins at room temperature. The supernatant was discarded, while the resulting pellet was resuspended in 2% (vol/vol) PTA in ddH$_2$O (pH 7.0) and 2% sarkosyl in DPBS. The samples were next incubated for one hour before the second centrifugation. The supernatant was removed and the pellet was re-suspended in DPBS. An aliquot of 5 µl of PTA purified brain homogenate was employed for electron microscopy (EM) analysis to confirm the presence of fibrils in each sample.

## Tau fibrillization and EM analysis

Synthetic human Tau protein (wildtype full-length monomers) was purchased as a lyophilized powder (rPeptide, T-1001–2) and resuspended in ddH$_2$O at a concentration of 2 mg/ml. The recombinant protein was fibrillized as described previously (*Guo and Lee, 2011*). Recombinant Tau was incubated with 40 µM low-molecular-weight heparin and 2 mM DTT in 100 mM sodium acetate buffer (pH 7.0) at 37°C, thereafter being agitated for seven days. The fibrillization mixture was centrifuged at 50,000 g for 30 mins, and the resulted pellet was resuspended in 100 mM sodium acetate buffer (pH 7.0) without heparin or DTT. Successful fibrillization was verified by EM.

Negative staining for EM analysis of fibrils was conducted as described elsewhere (*Eskandari-Sedighi et al., 2017*). Briefly, 400 mesh carbon-coated copper grids (Electron Microscopy Sciences) were glow-discharged for 40 s before adding the sample aliquots. PTA-purified brain homogenates or synthetic Tau fibrils (5 µL) were applied on the top of the grid for 1 min. These grids were washed

using 50 µl each of 0.1M and 0.01M ammonium acetate and negatively stained with 2 × 50 µl of filtered 2% uranyl acetate. After removing excess stain and drying, the grids were examined with a Tecnai G20 transmission electron microscope (FEI Company) with an acceleration voltage of 200 kV. Electron micrographs were recorded with an Eagle 4k × 4 k CCD camera (FEI Company).

## Liposome-mediated transduction of brain homogenate into tauopathy reporter cells

Polyclonal Tau4R-GFP cells were plated at $2 \times 10^5$ per well in 24-well plates. Cells were transduced the next day, using 40 µl of 10% clarified brain homogenate combined with Opti-MEM to a final volume of 50 µl. A further 48 µl of Opti-MEM and 2 µl of Lipofectamine-2000 (Invitrogen) was added to the previous Opti-MEM mixture to a total volume of 100 µl and incubated for 20 min. The liposome mixture was applied to the cells for 18 hr, and cells were then washed with PBS, trypsinized, and re-plated on coated coverslips (ThermoFisher) for imaging and analysis.

For PTA-precipitated brain homogenate, 1:10 dilution of precipitated fibrils was used for the transfection. 5 µl of PTA-purified fibrils was diluted in 45 µl Opti-MEM to a final volume of 50 µl. The previous Opti-MEM mixture was added to 47 µl of Opti-MEM and 3 µl of Lipofectamine-2000 and incubated in room temperature for 2 hr as described in *Safar et al., 1998*; *Woerman et al., 2016*. The mixture was added to cells, washed after 18 hr and re-plated before analysis exactly as mentioned previously.

## Quantification of the percentage of cells with positive inclusion

Prior to imaging, transfected cells were fixed 2% PFA in PBS for 15 mins. Samples were then washed twice with PBS then stained with DAPI (1:3000 from 1 mg/ml stock) for six mins. Cells were imaged using a Zeiss LSM 700 scanning confocal microscope featuring Zen 2010 software (Carl Zeiss, Oberkochen, Germany). Due to increased brightness of the GFP+ puncta formed after introduction of brain homogenate, GFP exposure was minimized for those cells only. To quantify the GFP+ puncta, a total of nine images were collected and analyzed for each condition, each with ~100 cells. DAPI-positive nuclei were utilized to determine the number of cells per image. The number of cells with inclusions (multiple nuclear inclusions or one cytoplasmic puncta) were counted and the percentage was calculated.

## Brain ventricle injections into tauopathy reporter larvae

Injections into the larval zebrafish brain (intraventricular space) were performed as described previously with few modifications (*Gutzman and Sive, 2009*). Embryos at 2 dpf (days post-fertilization) were removed from their chorions and anesthetized with 4% tricaine (MS-222, Sigma Aldrich). The embryos were placed in a 1% agarose-coated dish with small holes. Under a stereomicroscope, the immobilized embryos were oriented so that the brain ventricles were accessible for injections. The injection was carried out via pulled capillary tubes mounted in a micromanipulator. The injection volume was calibrated to 5 nL by injection into mineral oil and measurement with an ocular micrometer. Thereafter, the needle containing the injection solutions was placed through the roof plate of the hindbrain and 5–10 nL of either 10% clarified brain homogenate (TgTau$^{P301L}$ mice or wildtype littermate control), or synthetic tau, were mixed with 20% dextran Texas Red fluorescent dye (Invitrogen) and injected into the ventricles. For all the brain injection experiments, an uninjected control group and control group injected only with 20% red dextran fluorescent dye in PBS were included. After the injections, embryos were screened using a Leica M165 FC dissecting microscope and appropriately injected larvae were gathered for further analysis. The injections were considered appropriate if they had sharp edges and non-diffuse dye in the ventricle (*Figure 1—figure supplement 2*). Larvae receiving improper injections, in which the needle was inserted too deep in the brain ventricles resulting in the dye being visible outside the ventricle space and/or in the yolk, were excluded from analysis.

## Microscopy analysis of GFP-positive puncta in Tau reporter larvae

For the microscopic analysis of GFP-positive inclusions, larvae that were either injected or treated with traumatic injury, along with the control groups, were anesthetized via tricaine at the indicated time point (two, three, four, or five days post-injection (dpi) or post-traumatic injury (dpti) depending

on the experiment). Images for GFP-positive puncta on the brain area or lateral line above the spinal cord were taken using a Leica M165 FC dissecting microscope and the number of GFP-positive puncta were manually counted.

## TBI paradigm for zebrafish larvae

To induce TBI, 10–12 unanesthetized larvae (3 dpf) were loaded into a 10-ml syringe with 1 ml of E3 media. The syringe was blocked using a stopper valve to ensure no larvae or media left the syringe upon compression of the plunger. The syringe was held vertical using a metal tube holder at the bottom end of a 48' tube apparatus. A defined weight (between 30 and 300 g) was dropped manually from the top of the tube. The tube diameter was matched to (slightly greater than) the weight's diameter to enhance repeatability. This was either done once or repeated three times, with either 65 or 300 g weights. Once larvae were subjected to the TBI, they were moved back to a petri dish with fresh media and maintained for further analysis.

## Quantifying the pressure induced during TBI

To characterize the dynamic changes in pressure that occurred within the syringe during the TBI events, the stopper valve attached to the syringe (described immediately above) was replaced with a piezoresistive pressure transducer (#MLT844 AD Instruments, Colorado Springs, CO). Events were monitored via a PowerLab 2/26 data acquisition device and LabChart 7 software (AD Instruments). The pressure transducer was zeroed to report gauge pressure (pressure changes relative to atmospheric pressure) and was calibrated against a manometer (Fisherbrand Traceable from Thermoscientific, Ottawa ON). After each weight drop, the syringe apparatus was reset to remove any air bubbles and the pressure transducer was zeroed. Time courses of induced pressure were reported over a 350 msec time frame with 50 msec of base line recording, while mean and maximum pressure values were calculated from the initial 300 msec following the impact of the weight.

## Recording blood flow following TBI

Abnormalities of blood flow and circulation resulted from TBI was detected 5 to 10 mins after larvae were subjected to TBI. The blood flow in the tail area of zebrafish larvae, either those subjected to TBI or uninjured controls, was recorded using Leica DM2500 LED optical microscope.

## Measuring the Seizure-like phenotype in TBI larvae

The seizure-like behavior and activity of zebrafish larvae post-traumatic injury experiment was quantified via behavioral tracking software as described in our recent publications (*Kanyo et al., 2020a*; *Leighton et al., 2018*). Briefly, control larvae or larvae subjected to TBI using 65 g weight, were placed individually in wells of 96-well plates. The locomotor and seizure activity were assessed 40 min after the TBI through EthoVision XT-11.5 software (Noldus, Wageningen, Netherlands). The hypermotility of larvae is a manifestation of Stage I and Stage II seizures (previously defined via application of epileptic drugs), whereas more intense Stage III seizures are arrhythmic convulsions that manifest as reduced macroscopic movement in this assay (*Kanyo et al., 2020a*; *Leighton et al., 2018*; *Liu and Baraban, 2019*).

## Measuring neuronal activity during TBI using CaMPARI

We used a recently described (*Kanyo et al., 2020b*) in-house allele of transgenic zebrafish expressing the calcium sensor CaMPARI, line *Tg[elavl3:CaMPARI (W391F+V398L)][ua3144]*. Due to a federal moratorium on importing zebrafish into Canada (*Hanwell et al., 2016*), we remade these fish using the Tol2 transgenesis system (*Fisher et al., 2006*) and a vector gifted by Eric Schreiter's lab and published in *Fosque et al., 2015*. The transgene was bred onto the transparent Casper background.

Larvae with robust CNS expression of CaMPARI were loaded into a 20 ml syringe containing 1 ml E3 media (prepared as per recipe in *Westerfield, 2000*, but without ethylene blue) and were exposed to a 405 nm LED array (Loctite), which illuminated the syringe entirely. Larvae were exposed for 10 s, with the LED array at a distance of 7.5 cm from the syringe, while being subjected to TBI using the 300 g weight as described above. Following this photoconversion of CaMPARI during TBI, larvae were anesthetized in 0.24 mg/mL tricaine methanesulfonate (MS-222, Sigma Aldrich)

and embedded in 2% low-gelling agarose (A4018, Sigma Aldrich) for analysis under confocal microscopy.

CaMPARI imaging began with acquisition of Z-stacks (8 μm steps) using a laser point-scanning confocal microscope (Zeiss 700, 20x/0.8 Objective), and visualized as maximum intensity projections. The hindbrain area was analyzed, as it was the brain region most responsive to TBI. To specifically isolate the brain regions and obtain data points, a 3D area was isolated by creating a surface with Imaris 7.6 (Bitman, Zuerich) and the mean fluorescence intensities of the green and red channel intensities were calculated. Data points were presented as a red/green ratio for each individual larva and interpreted as relative neural activity, which is defined as red photoconverted CaMPARI in ratio to green CaMPARI (*Fosque et al., 2015*; *Kanyo et al., 2020b*).

## Bath application of drugs

Tau biosensor larvae were treated with 20 μM of the proteasome inhibitor MG-132 at 2dpf, following injections with brain homogenate from Tg human Tau mice. The treatment was left for 48 hr before changing the media and evaluating the percentage of larvae developing GFP+ puncta in the brain region.

For Kainic acid or kainate treatment (KA), the doses (5, 50, 100, 150, and 200 μM) were selected based on previous use of KA in zebrafish larvae (*Kim et al., 2010*; *Menezes et al., 2014*), and added within 6 hr after TBI. For 4-aminopyridine (4-AP), one of two doses of 4-AP (200 or 800 μM) were added either six or 24 hr after TBI, as indicated. For Retigabine (RTG) treatment, 10 μM was used to treat TBI larvae beginning 6 hr after TBI. Doses of 4-AP and RTG were selected based on our previous experience using them to affect seizures (*Kanyo et al., 2020b*). Unless otherwise stated, KA, 4-AP and/or RTG were applied to larvae for 38 hr, then a fresh drug-free E3 media was added. The formation of GFP-positive puncta was analyzed at four to five days post injury.

Pyrimidyn-7 (P7), the dynamin inhibitor, was purchased at a 50 mM concentration supplied in DMSO (Abcam). Larvae that were subjected to TBI were treated within six hours following the injury with 3 μM of P7. The dose was chosen based on the previous use of the P7 drug on zebrafish larvae (*Verweij et al., 2019*). The larvae were incubated with the drug for 20 hr, after which they were transferred to a fresh plate with drug-free media. Dyngo 4a, another dynamin inhibitor (*McCluskey et al., 2013*), was purchased from (Abcam) and 4 μM of Dyngo 4a was used to treat larvae as previously explained with P7. The formation and abundance of GFP-positive puncta was evaluated as previously described at 4 days post-traumatic injury (dpti). For some experiments, the 'Tau biosensor' transgenic zebrafish were bred to a separate Tg line that express human four repeat TAU Tg(*eno2:hsa.MAPT-ires-egfp*)[Pt406] throughout the zebrafish CNS (*Bai et al., 2007*).

## Statistics

All statistical analyses were performed using GraphPad Prism Software (Version 7, GraphPad, San Diego, CA). Sample sizes appropriate for our conclusions were estimated iteratively as the variance in each of our new methods became apparent; dose-response curves and significant differences amongst these dose were used to judge that any detected impacts of subsequent interventions would be valid. All experiments were independently replicated at least twice, individual larvae were the sampling unit (reported on Figures), and no outliers or other data were excluded. The experimenters were blinded to the treatments prior to quantifying outcomes. Paired t-tests were used to compare between two groups, except for when sample sizes were too small to assess normality wherein Mann-Whitney tests were used. For comparison between three or more groups at various time points or the same time point, two-way and ordinary one-way ANOVA were used followed by post-hoc Mann-Whitney U tests and Kruskal-Wallace multiple comparison tests, respectively.

## Acknowledgements

We acknowledge Nathalie Daude and David Westaway provided mouse brain samples, advice, and access to cell culture infrastructure. Gavin Neil and Jenna Bratvold contributed to SOD1-GFP cloning and zebrafish transgenesis via modifying a vector provided by Neil Cashman and Edward Pokrishevsky. Mark Loewen provided advice on hydraulic measures of pressure. Sue-Ann Mok, Satya Kar, Oksana Suchowersky, David Westaway and Brian Christie provided comments on an earlier version of the manuscript.

Funding to HA was from the Saudi Arabia Cultural Bureau and Majmaah University. RK was supported by SynAD postdoctoral fellowship funded via Alzheimer Society of Alberta and Northwest Territories through their Hope for Tomorrow program and the University Hospital Foundation. LFL received Studentships from Alberta Innovates and NSERC. MGD received Studentships from Alberta Innovates and CIHR. Operating funds to EB were from CurePSP 655-2018-06 and 468–08, U.S. Department of Veterans Affairs BX003168, and NIH NS080881; The contents of this article do not represent the views of the United States government. Operating funds to HW were from Alberta Innovates and the Alzheimer Society of Alberta and Northwest Territories through the joint Alberta Alzheimer's Research Program (AARP 201700005). Operating funds to WTA were also from the joint AARP (201700018), and from anonymous donors. Funders played no role in study design, prioritization, data collection or interpretation, or decision to submit the work for publication.

## Additional information

### Funding

| Funder | Grant reference number | Author |
|---|---|---|
| Alberta Innovates - Health Solutions | | Laszlo F Locskai<br>Michèle G DuVal |
| Natural Sciences and Engineering Research Council of Canada | | Laszlo F Locskai |
| Canadian Institutes of Health Research | | Michèle G DuVal |
| Alberta Innovates Bio Solutions | 201700005 | Holger Wille |
| Alberta Innovates Bio Solutions | 201700018 | W Ted Allison |
| Anonymous Donors | | W Ted Allison |
| National Institutes of Health | NS080881 | Edward A Burton |
| CurePSP | 655-2018-06 and 468-08 | Edward A Burton |
| U.S. Department of Veterans Affairs | BX003168 | Edward A Burton |
| Majmaah University | | Hadeel Alyenbaawi |
| Saudi Arabia Cultural Bureau in Ottawa | | Hadeel Alyenbaawi |

The funders had no role in study design, data collection and interpretation, or the decision to submit the work for publication.

### Author contributions

Hadeel Alyenbaawi, Conceptualization, Resources, Formal analysis, Supervision, Funding acquisition, Investigation, Visualization, Methodology, Writing - original draft, Project administration, Writing - review and editing; Richard Kanyo, Formal analysis, Supervision, Investigation, Visualization, Methodology, Writing - review and editing; Laszlo F Locskai, Resources, Formal analysis, Supervision, Investigation, Visualization, Methodology, Writing - review and editing; Razieh Kamali-Jamil, Investigation, Writing - review and editing; Michèle G DuVal, Qing Bai, Resources, Investigation, Writing - review and editing; Holger Wille, Edward A Burton, Resources, Supervision, Writing - review and editing; W Ted Allison, Conceptualization, Resources, Formal analysis, Supervision, Funding acquisition, Investigation, Visualization, Methodology, Project administration, Writing - review and editing

### Author ORCIDs

Michèle G DuVal  http://orcid.org/0000-0001-6975-8117
Holger Wille  http://orcid.org/0000-0001-5102-8706
Edward A Burton  http://orcid.org/0000-0002-8072-4636
W Ted Allison  https://orcid.org/0000-0002-8461-4864

## Ethics

Animal experimentation: Our Discussion invokes animal research ethics as an advantageous aspect of our technology development. Our Methods section begins with the following: Animal Ethics and Zebrafish Husbandry Zebrafish were raised and maintained following protocol AUP00000077 approved by the Animal Care and Use Committee: Biosciences at the University of Alberta, operating under the guidelines of the Canadian Council of Animal Care.

## Decision letter and Author response

Decision letter https://doi.org/10.7554/eLife.58744.sa1
Author response https://doi.org/10.7554/eLife.58744.sa2

## Additional files

### Supplementary files

• Source data 1. Source data and statistics.

• Transparent reporting form

### Data availability

All data generated or analysed during this study are included in the manuscript and supporting files. Source data files have been provided in Source Data 1.

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
