## [Decision Letter]

**Acceptance summary:**

Your manuscript provides a new model for this form of brain injury. The accessibility of the work in larval zebrafish will enable a powerful combination of environmental, genetic and imaging tools to learn more about this pathogenic process – and to hopefully find new ways to treat any of us that are unfortunate to suffer from this kind of trauma.

**Decision letter after peer review:**

Thank you for submitting your article "Seizures are a druggable mechanistic link between TBI and subsequent tauopathy" for consideration by *eLife*. Your article has been reviewed by three peer reviewers, including Stephen C Ekker as the Reviewing Editor and Reviewer #1, and the evaluation has been overseen by Didier Stainier as the Senior Editor.

The reviewers have discussed the reviews with one another and the Reviewing Editor has drafted this decision to help you prepare a revised submission.

Summary:

The authors present a well-designed study to evaluate the validity of a larval model of traumatic brain injury in zebrafish, a biosensor to report on the prevalence of Tau aggregation and subsequent studies aimed at delineating some of the mechanistic pathways linked to the prion-like progression of tauopathy and relationship to post-traumatic epilepsy. They conclude with a demonstration test of anticonvulsants as a potential example intervention. The authors should be commended on their simple model and extensive body of work that has gone into validating the biosensor and its application in a zebrafish model. Moreover, this body of work further validates the use of zebrafish as an important tool and bridge organism in the field of neurotrauma.

The work is also well described with suitable experimental study design to minimize bias and provides data for full transparency.

We note a number of unresolved questions based on the body of work in the existing manuscript. We note these as opportunities for improvement through textual clarification. Should you have additional data on-hand to add, that is also appropriate but not required.

Essential revisions:

1) Acknowledging the previous work published by McCutcheon et al., 2016 that reported an excitotoxic model of TBI with potential for high throughput drug screening in TBI.

2) Please provide a description and comparison of their work with recent data in an adult zebrafish model of TBI by Cho et al., 2020.

3) Although the Materials and methods section of the manuscript is quite detailed, the Results section sometimes lacks experimental information. Some examples are itemized below.

Points for Improvement:

1) Figure 1G-I. The differences between fish injected with the TG brain (experimental) and controls is not large. Moreover, the presence of aggregates in control is surprising. Furthermore, such aggregates are less prominent in the controls of the experiment with the human protein (Figure 2C-E) compared to the Tg mouse extract (Figure 1). Finally, the spontaneous aggregates in the spinal cord of fish in the experiment shown in Figure 5 is another surprising observation. This is discussed in the subsection “ Limitations of our approach” but the matter remains somewhat problematic.

2) Why the increase in neural activity after TBI are particularly prominent in the hindbrain is something that could be discussed.

3) In the first section of their Discussion, the authors state that their "zebrafish models imperfectly replicate human TBI and tauopathy…". It would have been helpful to the reader if they had first summed up why they thin this replication is imperfect. This comment has to do with the general organization of the Discussion and it could have preferable for the authors to start with the third section of the Discussion and perhaps introduce some elements found in subsection “ Limitations of our approach”. I leave it to the author's judgment.

4) Subsection “Bath Application of Drugs”: How were these doses determined to be appropriate? Reference?

5) Figure 1 panels E and F: figure legend should indicate when these microphotographs were taken after application or injection, respectively.

6) Subsection “Validation of in vivo Tau biosensor via intra-ventricular brain injections of Tau fibrils”: The shape of the puncta. Can the authors refer to a specific figure or provide a higher magnification of a figure panel.

7) Subsection “Validation of in vivo Tau biosensor via intra-ventricular brain injections of Tau fibrils” end of paragraph three: The text is redundant and reference to Figure 2A is not really appropriate. Some minor rewriting is necessary.

8) Many figures are lacking scale bars.

9) Paragraph four of subsection “Introduction of the first traumatic brain injury model for larval zebrafish”: It should be Figure 3F and not 3B. Furthermore, the signal is hard to see in the right half of the Figure.

10) The red color images in panels B' and B' of Figure 4 are not particularly convincing, at least on my PDF version of the manuscript.

11) The mass that was used for the experiment shown in Figure 5A-D should be indicated.

12) Discussion paragraph four: The authors specify what a "blast injury" is: "(e.g as experienced by military.….)". However, this is at least the third time the term is mentioned. Perhaps move the explanation to the first occurrence.

13) Typos, grammar: Subsection “Recording blood flow following TBI” "were subjected"

14) "data" is plural. Please correct all instances of "data is" to "data are", "this data" to "these data" throughout the manuscript.

15) It is not entirely clear in the manuscript why the SOD-GFP transgenic was an appropriate control for comparison to the Tau-GFP biosensor in terms of aggregation after trauma. Please elaborate.

16) The model is simple to apply but raises some concerns about the interpretation of data and specific relevance to TBI. For example, the data in Figure 3 examines hemorrhage and blood flow impairments after trauma, however, the method of injury application is likely to extend injury to the entire larvae and not only to the head. This is highlighted in the data in Figure 3 that demonstrates blood flow impairments in the tail and hemorrhage in areas outside of the brain region. (It is difficult to discern whether the apoptotic cells were superficial, within the brain region specifically, or whether other parts of the fish were also demonstrating apoptotic markers (Figure 3—figure supplement 1)). The authors should acknowledge the limitations of this particular injury paradigm and the interpretation of the data.

17) The authors should provide some explanation as to why tau aggregation in the brain appears limited to a distinct region in the midline and specific somite regions within the spinal cord.

---

## [Author Response]

Essential revisions:1) Acknowledging the previous work published by McCutcheon et al., 2016 that reported an excitotoxic model of TBI with potential for high throughput drug screening in TBI.

We now acknowledge this excitotoxicity model in our Results.

2) Please provide a description and comparison of their work with recent data in an adult zebrafish model of TBI by Cho et al., 2020.

Thank you for directing us to this relevant work. We have now briefly contrasted this work with our larval approach in the Discussion.

3) Although the Materials and methods section of the manuscript is quite detailed, the Results section sometimes lacks experimental information. Some examples are itemized below.

We address each point in turn below.

Points for Improvement:1) Figure 1 G-I. The differences between fish injected with the TG brain (experimental) and controls is not large.

The purpose of Figures 1 and 2 is to validate our new Tau-GFP biosensor line, showing that we can detect Tau aggregates after an intervention known to provoke Tau aggregation. The differences shown are > 3-fold and statistically significant. We do not doubt that further optimization of Tau fibril injections in this model could increase the effect size, but this is not the objective of the present work.

Moreover, the presence of aggregates in control is surprising.

The detection of occasional aggregates in control zebrafish might reflect real stochastic events that are not observable in other models, since this is the first in vivo system allowing observation of Tau aggregation. It is conceivable that these events reflect the proteostatic concepts of dynamic balance between accumulation and clearance of misfolded proteins, in a rapidly developing CNS where some cells may express higher levels of Tau or the reporter gene. However, even if the detection of occasional aggregates in control zebrafish is a transient technical artifact, it is not “somewhat problematic”. All data are reported as comparative differences between experimental groups and controls. Figures 1, 2 and 5 show unequivocally that the number of GFP puncta was increased relative to controls after injection of transgenic mouse brain homogenate, synthetic fibrils, exposure to proteasome inhibitors or fluid shock wave injury, but not by control mouse brain homogenate, vehicle, Tau monomer, or sham injury. The reasonable and parsimonious interpretation is that the reporter shows Tau aggregates, allowing us to use this a tool to interrogate the mechanisms underlying tauopathy in TBI.

Furthermore, such aggregates are less prominent in the controls of the experiment with the human protein (Figure 2 C-E) compared to the Tg mouse extract (Figure 1). Finally, the spontaneous aggregates in the spinal cord of fish in the experiment shown in Figure 5 is another surprising observation. This is discussed in the subsection “ Limitations of our approach” but the matter remains somewhat problematic.

Minor quantitative differences between controls in replicate experiments are routinely seen in all biological studies, regardless of discipline, and are unavoidable because of the large number of undefined variables in complex biological systems. This is why all of our experiments include controls from the same clutch of embryos that replicate genetics, epigenetics, and environmental conditions exactly, so that any differences in the number of aggregates are unequivocally attributable to the experimental intervention. This is also why all of the experiments were carried out in biological triplicate, an advantage of using zebrafish models whose throughput allows this degree of experimental rigor.

2) Why the increase in neural activity after TBI are particularly prominent in the hindbrain is something that could be discussed.

We added some speculation on this point to the Discussion section that considers the link of post-traumatic seizures to subsequent tauopathy.

3) In the first section of their Discussion, the authors state that their "zebrafish models imperfectly replicate human TBI and tauopathy…". It would have been helpful to the reader if they had first summed up why they thin this replication is imperfect. This comment has to do with the general organization of the Discussion and it could have preferable for the authors to start with the 3rd section of the Discussion and perhaps introduce some elements found in subsection “ Limitations of our approach”. I leave it to the author's judgment.

We broadly have left the order of the information as originally submitted, and tried to partially fix this issue by adjusting the writing. We now indicate for the reader that the limitations (the “imperfect” part) will be discussed in a subsequent section.

4) Subsection “Bath Application of Drugs”: How were these doses determined to be appropriate? Reference?

We removed the word “appropriate” from this sentence and added references to the literature that guided our dose selection, including two papers that previously reported seizure responses to Kainate in zebrafish larvae. Please also note we established a dose-response curve using this drug (Figure 5—figure supplement 5).

5) Figure 1 panels E and F: figure legend should indicate when these microphotographs were taken after application or injection, respectively.

We have added a sentence clarifying this at the end of the legend to Figure 1.

6) Subsection “Validation of in vivo Tau biosensor via intra-ventricular brain injections of Tau fibrils”: The shape of the puncta. Can the authors refer to a specific figure or provide a higher magnification of a figure panel.

We have made this description less definitive, reflecting our perception that the puncta have various characters, and refer the reader to Figure 1F”.

7) Subsection “Validation of in vivo Tau biosensor via intra-ventricular brain injections of Tau fibrils” end of paragraph three: The text is redundant and reference to Figure 2A is not really appropriate. Some minor rewriting is necessary.

We have adjusted this writing.

8) Many figures are lacking scale bars.

We have added scale bars to the remaining figure panels.

9) Paragraph four of subsection “Introduction of the first traumatic brain injury model for larval zebrafish”: It should be Figure 3F and not 3B. Furthermore, the signal is hard to see in the right half of the Figure.

We have corrected this to reference to panel 3F. We have now changed the pseudocolored panel from magenta to grayscale to increase the visibility of the caspase-3 labeled cells, and added arrows to assist. Our quantification of the signals was performed while viewing the intact larvae in a fluorescent dissection scope (where the signal is obvious), rather than from these types of images. The observers were blinded to the treatments. Further data convincing the results are consistent and robust include the quantification of a few dozen larvae in a separate experiment performed by separate observers (Figure 5H).

10) The red color images in panels B' and B' of Figure 4 are not particularly convincing, at least on my PDF version of the manuscript.

Thank-you. We have adjusted the single-channel images to grayscale, and please note the red is pseudocoloured to magenta in the merged panels. In our view, the abundance of red/magenta is convincingly greater in B’ compared B”. One should not expect the difference between this to be large (as it is when we use the same CaMPARI quantification in other paradigms, e.g. Kanyo et al., 2020), because here the time window of photo-conversion was intentionally kept very brief. Regardless, this is a robust and reliable signal, that includes both an increase in red fluorescence and concomitant decrease in green fluorescence as the CaMPARI is photoconverted. That alteration is best appreciated from its quantification of ratiometric red:green CaMPARI fluorescence – it is readily obvious when that ratio is reported in the heatmaps of panel C, or when quantified (from 3D volumetric renderings) to show it is quite consistent between trials in panel D.

11) The mass that was used for the experiment shown in Figure 5A-D should be indicated.

We have added this to the beginning of the figure legend.

12) Discussion paragraph four: The authors specify what a "blast injury" is: "(e.g as experienced by military.….)". However, this is at least the third time the term is mentioned. Perhaps move the explanation to the first occurrence.

Thank-you. We have added a phrase to our first use of “blast injury”. Based on our experience

from other presentations of this work we think it is helpful to remind the reader of this mode of TBI. For example some readers will be more accustomed to rodent TBI models that typically implement a piston to deliver a direct physical blow. We feel it is worthwhile to remind the reader that while our model is different than (and complements) these approaches, our model is nonetheless representative of a mode of injury that important in human TBI.

13) Typos, grammar: Subsection “Recording blood flow following TBI” "were subjected"

Corrected.

14) "data" is plural. Please correct all instances of "data is" to "data are", "this data" to "these data" throughout the manuscript.

We have corrected every instance of incorrect usage we found.

15) It is not entirely clear in the manuscript why the SOD-GFP transgenic was an appropriate control for comparison to the Tau-GFP biosensor in terms of aggregation after trauma. Please elaborate.

SOD1-GFP was selected because it is a similar GFP-based biosensor that is designed to report protein aggregation, and because SOD1 aggregation is not thought to be involved in the response to TBI. The absence of aggregates in SOD1-GFP zebrafish after TBI provides unequivocal proof that the effect seen for the Tau biosensor is specific to Tau. This additional control enhances the degree of scientific rigor in our study. We now mention this rationale more clearly at first occurrence in the Results.

16) The model is simple to apply but raises some concerns about the interpretation of data and specific relevance to TBI. For example, the data in Figure 3 examines hemorrhage and blood flow impairments after trauma, however, the method of injury application is likely to extend injury to the entire larvae and not only to the head. This is highlighted in the data in Figure 3 that demonstrates blood flow impairments in the tail and hemorrhage in areas outside of the brain region. (It is difficult to discern whether the apoptotic cells were superficial, within the brain region specifically, or whether other parts of the fish were also demonstrating apoptotic markers (Figure 3—figure supplement 1)). The authors should acknowledge the limitations of this particular injury paradigm and the interpretation of the data.

We agree with this important point, and have now alluded to this at the beginning of the Discussion (where we describe “blast injuries” and added a sentence to the section “Limitations of our approach” next to our description of difficulties in appreciating the physical events that are happening in the larvae. We respectfully point out that the same is true of human blast injuries, which often cause brain trauma as a component of a wider clinical picture, so this may enhance the translational relevance of the model.

17) The authors should provide some explanation as to why tau aggregation in the brain appears limited to a distinct region in the midline and specific somite regions within the spinal cord.

We added some speculation on this point to the Discussion section that considers the link of post-traumatic seizures to subsequent tauopathy.